# Inflammatory biomarkers and perinatal depression: A systematic review

**Anabela Silva-Fernandes**[1][☯], **Ana Conde**[2][☯], **Margarida Marques**[3], **Rafael A. Caparros-Gonzalez**[4,5], **Emma Fransson**[6], **Ana Raquel Mesquita**[1,7], **Bárbara Figueiredo**[1], **Alkistis Skalkidou**[6]*

**1** Psychology Research Center (CIPsi), School of Psychology, University of Minho, Braga, Portugal, **2** CINTESIS@RISE, CINTESIS.UPT, Portucalense University, Porto, Portugal, **3** Faculdade de Psicologia, CICPSI, Universidade de Lisboa, Lisboa, Portugal, **4** Faculty of Health Sciences, Department of Nursing, University of Granada, Granada, Spain, **5** Instituto de Investigación Biosanitaria ibs, GRANADA, Granada, Spain, **6** Department of Women's and Children's Health, Uppsala University, Uppsala, Sweden, **7** ProChild CoLab Against Poverty and Social Exclusion–Association (ProChild CoLAB) Campus de Couros Rua de Vila Flor, Guimarães, Portugal

☯ These authors contributed equally to this work.

* alkistis.skalkidou@uu.se

**Data Availability Statement:** All relevant data are within the manuscript and its Supporting Information files.

## Abstract

### Background

Approximately 10 to 20% of pregnant women worldwide experience perinatal depression (PND), a depressive episode with onset during pregnancy or after childbirth. We performed a systematic review to identify, summarize and discuss studies on inflammatory biomarkers described in relation to PND.

### Method

Inclusion criteria defined the selection of observational studies written in English, French, Spanish or Portuguese, that evaluate analytical levels of inflammatory molecules (protein levels) in biological fluids in women, with a diagnosis of depression using ICD/DSM diagnostic criteria or depressive symptoms assessed by standardized psychometric instruments, during pregnancy and/or postpartum. Case reports, experimental studies, reviews, qualitative analysis, meta-analysis, gray literature or replicated data were excluded. Three electronic databases were used for search (Pubmed, Web of Science and PsychInfo) and quality assessment of selected studies were performed using the Newcastle-Ottawa Scale. Data extraction included study design; number of subjects; obstetric information; tools and timepoints of depression and inflammatory markers assessment.

### Results

56 studies (sample size for cross-sectional and case-control studies ranging from 10 to 469; sample size for longitudinal studies ranging from 26 to 467), where the major aim was to analyze the association between depression and inflammatory biomarkers during pregnancy and postpartum period were included in this systematic review. Overall, the findings of our systematic review lend support to the hypothesis that several inflammatory markers may be associated with peripartum depressive symptoms. The associations were somewhat

**Funding:** AS-F was supported by FCT and the Portuguese Ministry of Science, Technology and Higher Education, through national funds, within the scope of the Transitory Disposition of Decree No. 57/2016, of 29th of August, amended by Law No. 57/2017 of 19 July and her work is conducted at the Psychology Research Centre (CIPsi), School of Psychology, University of Minho, supported by FCT through the Portuguese State Budget (Ref.: UIDB/PSI/01662/2020). AS was supported by the Uppsala Region ALF funds. AM was supported from the Portuguese Foundation for Science and Technology (FCT) and from EU through the European Social Fund and from the Human Potential Operational Program - IF/00750/2015. Psychology Research Centre (CIPsi), School of Psychology, University of Minho, supported by FCT through the Portuguese State Budget (Ref.: UIDB/PSI/01662/2020). ProChildCoLAB is supported by Mission Interface Program from the Resilience and Recuperation Plan, notice n° 01/C05-i02 /2022, approved by ANI - Agência Nacional de Inovação, S.A. BF was supported from the Portuguese Foundation for Science and Technology (FCT) and from EU through the European Social Fund and from the Human Potential Operational Program - IF/00750/2015. Psychology Research Centre (CIPsi), School of Psychology, University of Minho, supported by FCT through the Portuguese State Budget (Ref.: UIDB/PSI/01662/2020). This paper is part of the COST Action Riseup-PPD CA18138 and was supported by COST under COST Action Riseup-PPD CA18138 (Virtual Mobility Grant). The funders had no role in study design, data collection and analysis, decision to publish, or preparation of the manuscript.

**Competing interests:** The authors have declared that no competing interests exist.

different looking at pregnancy compared to the delivery time-point and postpartum, and mainly referred to increased levels of IL-6, IL-8, CRP and TNF-α among depressed.

## Discussion

In summary, our systematic review findings provide evidence supporting the hypothesis that several inflammatory markers may correlate with peripartum depressive symptoms. However, our work also highlighted notable differences in the timing of biological sampling for inflammatory markers and in the methodologies used to assess depression during the perinatal period. Additionally, variations were observed in how inflammatory biomarkers and depression were approached, including their classification as exposure or outcome variables, and the timing of assessments. It is essential for future research to investigate the influence of biological fluids and the timing of assessments for both inflammatory biomarkers and depression to gain a deeper understanding of their association. This comprehensive exploration is pivotal for elucidating the intricate relationship between inflammation and perinatal depression.

## Introduction

Pregnancy and postpartum are critical periods for the mental health of the mother, her baby and the whole family. During the past decades, knowledge regarding the psychobiological pathways impacting on mental health has expanded substantially, including studies in the perinatal setting. Increasing evidence supports the link between psychosocial and biological pathways, especially on the role of the immune system in the development of perinatal depression (PND), both with antenatal (AND) and postpartum onset (PPD) (e.g., [1–4]).

The immune system is a complex network that aims to protect the host from invading microorganisms and induce wound healing when needed. During an immune response, human behavior is affected, leading often to increased inactivity and sleepiness, decreased appetite and social withdrawal, behaviors that also resemble those characteristic of clinical depression [5,6]. The interplay between the immune functioning and depression has been explored over the last decades and a bidirectional loop has been described. While inflammation seems to play a key role in depression's pathogenesis, at least for a subset of depressed individuals, it was also been shown that depression, adversity and stress have also been associated to exaggerated or prolonged inflammatory responses (for a review see [7]). Suggested mechanisms include increased inflammation peripherally increasing inflammation in the brain [6,8] through increased permeability of the blood-brain barrier (BBB) [9] and increased indoleamnie-pyrrole 2,3-dioxygenase (IDO) activity leading to decreased serotonin synthesis, and thus contributing to depression [10–12].

The dramatic shift in the characteristics of the immune response during pregnancy and postpartum seem to impact on maternal pregnancy mood [13]. The increase in regulatory T-cells (Treg) occurring in mid-pregnancy coincide with the period when most women report increased well-being, and Treg activity has been associated with resilience to stress in animal studies [14]. The late third trimester and the delivery itself could be characterized as a largely pro-inflammatory periods [15,16]. Intensification of pro-inflammatory activity in late pregnancy co-occurs with an increase in depressive symptoms during this period [17].

After childbirth, the body needs to reduce pro-inflammatory activity, and a decrease of many inflammatory markers has been noted from the third trimester to the postpartum

[18,19]. For many women, this period is characterized by additional bodily changes associated with wound healing and breast-feeding onset that could also impact on immune actions. Furthermore, sleep disturbances that are common perinatally, are also known to induce a pro-inflammatory state [20] affecting depression risk. The early postpartum period has been characterized by a drop in Treg, and an immune response characterized by T helper 1 (Th1) or Macrophages type 1 has been described [21]. While these changes in the immune system might be universal for the postpartum period, they are associated with an increased risk of depression during the postpartum period in some individuals (e.g., [4,22,23]). Why only some individuals experience mood symptoms in the perinatal period might be due to differences in the sensitivity to those changes in both hormonal as well as inflammatory marker levels. Studies have also suggested that dysregulation of cytokine production may contribute to mood disorders in the perinatal period (reviewed in [24–26]). It is interesting that the level of the inflammatory response across pregnancy and childbirth could vary according to personality [27], mental health during pregnancy [28] and previous traumatic exposures [29].

Several reviews have been already conducted aiming to explore immune system functioning and the role of inflammation in depression during pregnancy [24]. A previous systematic review on postpartum depression included 25 articles [30]. Their most robust finding was that levels of CRP in late pregnancy could predict postpartum depression. A substantial increase in the literature in this domain was observed during the past few years, and many new studies on this topic have been published, especially with more focus to depression with antenatal onset. In parallel, new techniques for biomarker analyses have developed further. A recent meta-analysis of inflammatory markers for major depression, showed robust results of increase of several pro-inflammatory markers in depressed individuals [31]. Nevertheless, some studies on other specific inflammatory biomarkers in the perinatal setting show conflicting results. A recent review of literature has suggested that distinct and changing inflammatory profiles throughout pregnancy and postpartum could exist, which makes the study of depression-related alterations in these periods highly complex [13]. Another very recent systematic review [32] alerts in relation to the lack of clarity regarding a consistent immune profile, especially based on the analysis of circulating peripheral cytokines. In fact, despite a significant number of studies assessing potential immunological markers of perinatal depression, it does not appear that levels of any individual pro- or anti-inflammatory marker is a useful predictor of perinatal depression, especially considering the evidence for interactions between depression and maternal psychosocial factors [32,33]. Despite the growing evidence in the field of immune related biomarkers in PND and PPD, clinical applications for biomarkers for depression prediction or treatment during these periods are lacking in clinical practice.

Our systematic review endeavors to thoroughly investigate the relationship between inflammation and depression throughout the perinatal period. This includes an extended and in-depth analysis of studies exploring how this association varies depending on when it is assessed (pregnancy, postpartum or from pregnancy to postpartum). Furthermore, we aim to delve into longitudinal studies to not only investigate cross-sectional between inflammation and perinatal depression but also to elucidate possible bidirectional effects between them. By comprehensively examining these factors, we aim to provide valuable insights into the complex interplay between inflammation and perinatal depression, contributing to a better understanding of this critical aspect of maternal and infant health.

## Methods

This study followed the recommendations outlined in the Preferred Reporting Items for Systematic Review and Meta-Analysis (PRISMA) guidelines [34,35] and has been registered on

PROSPERO (Registration ID: CRD42020210080). Protocol details are available at https://www.crd.york.ac.uk/PROSPERO/display_record.php?RecordID=210080.

## Search strategy

An initial article search was conducted on July 3rd, 2020, through three electronic databases (Pubmed, Web of Science and PsychInfo) to identify English, Portuguese, French and Spanish-language human studies unrestricted by year of publication (for search terms see S1 Table). On February 23rd, 2022, a new search was conducted using the same search formula and filters (except for the Filter "Journal Article" in Pubmed that is no longer available) to determine new entries. After duplicates removal, the unique entries from this new search were identified by table comparison.

Duplicate detection was performed by two independent reviewers (ASF and MM) on two different platforms (Endnote and Rayyan) using manual review by ordering articles by title, authors, pages, journal. Finally, the authors met to reach accordance on the final number of duplicates.

## Studies selection

Original quantitative studies that evaluate the levels of inflammatory molecules in women with a diagnosis of depression or depressive symptomatology conducted in women during pregnancy and/or at the postpartum period (up to one year after delivery) were eligible for this systematic review. Two authors (ASF and MM) screened the titles and abstracts of articles from the primary search independently against inclusion and exclusion criteria:

Inclusion criteria:

- Written in English or French or Spanish or Portuguese

- Observational studies

- Depression assessed using ICD/DSM diagnostic criteria either through diagnostic interview or expert opinion. Alternatively, depressive symptomatology assessed using standardized psychometric instruments

- Inflammatory molecules measurement (protein levels) in biologic fluids using analytical techniques

Exclusion criteria:

- Case reports or experimental studies or reviews or qualitative analysis or meta-analysis or gray literature

- Replicated data

The full text of qualifying articles was then assessed against the same standard by different pairs of authors (ASF&MS; AC&ARM; BF&RCG; AS&MM). Any discrepancies were resolved first through discussion amongst the pair, and if a consensus could still not be reached, by conferring with other group members.

## Data extraction

The following data was extracted from each selected study: country of origin; study design; number of subjects; socio-economic status/ethnicity; obstetric information, namely delivery mode; assessment of depression: instrument(s) and timepoint(s) and inflammatory protein markers: biological fluid, hour of collection, timepoint(s), dosage assessment technique, inflammatory markers and results.

## Quality assessment

Following PRISMA guidelines, the quality assessment of selected studies (RCG&MM) and the data extraction were conducted independently by two authors. At the end, the complete data extraction table was revised to uniformization. Different versions of the Newcastle-Ottawa Scale (NOS) were used to assess the methodological quality of selected studies, namely case-control and cohort studies [36,37]. A 'star system' has been developed in which a study is judged on three broad perspectives: the selection of the study groups; the comparability of the groups; and the ascertainment of either the exposure or outcome of interest for case-control or cohort studies respectively with a maximum score of 9 points [37]. An adapted form of NOS for cohort studies was used for quality assessment of the cross-sectional studies with a maximum score of 10 points [38]. All the inter-rater agreements between authors were verified prior to resolving disagreements.

## Results

A total of 3527 relevant references were initially identified in an electronic search of three databases: Pubmed, Web of Science and PsychInfo. All 806 duplicated references were removed, and 2721 articles remained. The titles and abstracts of the identified references were screened, and 2594 non-relevant references were excluded. The full text of the 127 remaining studies was then screened, and 44 studies met one or more exclusion criteria. At the final stage, 83 studies were included in the review. A flow diagram of the search selection for the included studies is presented in Fig 1.

From the 83 studies included in the qualitative analysis 27 were studies in which the association between depression and inflammatory markers wasn´t the principal aim but since this derived data was available, we performed data extraction from these studies and presented them in the S2 Table.

For the accomplishment of the aims of this systematic review we have focused on the 56 studies where the major aim was to analyze the association between depression and inflammatory biomarkers during pregnancy and the postpartum period. Considering the study design used, 8 studies (3 of which with prospective analysis) used a case-control design, 20 studies a cross-sectional, and 27 a cohort analysis design. Thirty-one studies involved repeated assessment time points and longitudinal analyses both during pregnancy (4 studies) and from pregnancy to postpartum (27 studies).

The studies included originate from different countries and research settings, while the involved participants present with markedly different sociodemographic and clinical features (e.g., with and without psychosocial risks). Further, depression was assessed with different methods, from self-reported questionnaires for assessing depressive symptoms, depressive mood, depressive symptomatology or clinical interviews for diagnosing depressive disorders. Lastly, inflammation-related molecules were assessed with very different techniques and in several biological fluids, mostly in serum (30 studies) and plasma (22 studies), but also in blood (one study), breast milk (two studies), urine (one study), CSF (two studies) and PBMC cells (two studies).

Due to this diversity, results are presented in two steps. Firstly, cross-sectional, and case-control studies on inflammatory biomarkers and depression are presented according to the period involved (pregnancy or postpartum) (for details see Table 1) involving a sample size range between 10–469. Secondly, longitudinal studies are presented, with a sample size ranging from 26 to 467, with a note on whether the original study included predictive analyses (for details see Table 2). A summary table is provided as S3 Table.

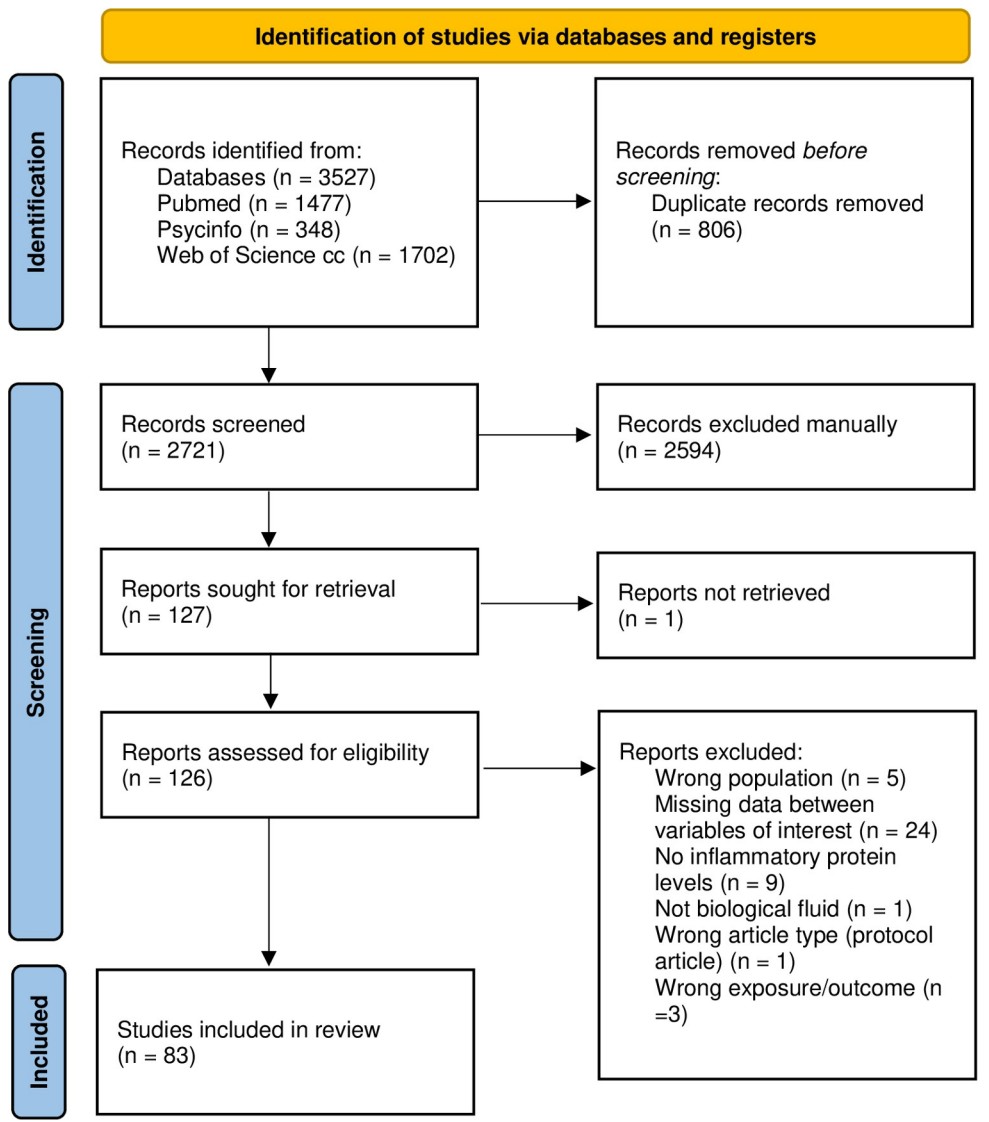

**Fig 1. Flow diagram of the search selection for the included studies.**

## Quality assessment

Two researchers analyzed 56 articles for quality assessment. The inter-rater reliability between authors was analyzed calculating the agreement percentage (84%) and Cohen's kappa (K = 0.777), significance p-value < .001. In general, independently of the study design, studies with high quality (higher or equal to 8 points) are scarce. As can be seen in Table 1, the quality of the fourteen studies (4 case-control and 10 cross-sectional) that have assessed the association between depression and inflammatory biomarkers during pregnancy ranged between 6 and 8 points (M = 6.54; SD = 0.66); and the eleven studies assessing this association during postpartum ranging between 2 and 9 points (M = 6.73; SD = 1.26), with a study without quality assessment due to lack of information (conference paper). Regarding the longitudinal studies, the quality of the four studies that have been implemented during pregnancy ranged between 5 and 8 points (M = 6.5; SD = 1.00), the 22 studies developed between pregnancy and postpartum ranged between 4 and 8 points (M = 6.27; SD = 0.84) and the five studies that were developed during the postpartum period have ranged between 6 and 8 points (M = 6.4; SD = 0.64).

**Table 1. Cross-sectional and case-control studies on inflammatory biomarkers and depression.**

| Reference | Country | Study design | Number of subjects | Socio-economic status/ethnicity | Delivery mode | Instruments | Timepoint(s) | Biological fluid/hour of collection | Timepoint(s) | Dosage assessment technique | Inflammatory markers | Results | Total quality score |
|---|---|---|---|---|---|---|---|---|---|---|---|---|---|
| | | | | | | | | | Inflammatory protein markers | | | | |
| | | | | | | Assessment of depression | | | | | | | |
| **Pregnancy** | | | | | | | | | | | | | |
| [39] | USA | Nested case-control (from a prospective cohort study) | T = 462 (Ct = 390, PND = 72) | 39.5% college graduate; 40.7% non-white race; | NR | Perinatal depression documented by the obstetric or the primary care provider in the electronic medical record (EMR) of the study institution. | Anytime across pregnancy | Plasma/NR | Average of three visits: 10, 18 and 26 gw | (1) Hs Multiplex Assay; (2) DuoSet ELISA | (1) IL-6, IL-10, CRP, IL-1β, TNF-α, (2) CRP | NS | 6 |
| [40] | USA | Cross-sectional | T = 187 | 87.2% ≥High school education; 87.7% reported income of $34,315.2±31,431.2; 100% African American | NR | Depressive symptoms measured as CES-D total score; CES-D ≥ 16 equate with symptoms of depression; | 13.1–28.6 gw | Serum/NR | 13.1–28.6 gw | (1) ELISA; (2) custom 4-plex assay | (1) hs-CRP, (2) IL-1β, IL-6, IL-10 and TNF-α | (+) correlation of CES-D with IL-6 (p =.04) and IL-1β (p =.03) | 7 |
| [41] | Finland | Cross-sectional (from a cohort study) | T = 139 | 5.8% low education levels | NR | EPDS: continuous total sum score was used for the main analyses. Additional comparisons (high/low EPDS) with the cut-point 9/10. | 24 gw | Serum/randomly through the office hours | 24 gw | Bio-Plex Pro Human Cytokine 21- and 27-Plex Assay kits | IL-6, IL-10, TNF-α, IL-9, IL-5, IL-12, IL-13, IFN- γ, IL-4 | (+) correlations between EPDS scores and four cytokines: IL-5 (p =.007), IL-9 (p =.011), IL-12 (p =.018), IL-13 (p =.029) and the IFN-γ/IL-4 ratio (p =.039). | 6 |
| [42] | USA | Cross-sectional | T = 60 | 82% high school or less education; 63% annual family income < $15,000; 57% African American | NR | Depressive symptoms measured as CES-D total score (clinical cut-off 16) | M = 15±7.8 gw | Serum/9:30 am-1:30 pm | 15±7.8 gw | Ultra sensitive multiplex kits | IL-6, TNF-α | NS | 7 |
| [43] | USA | Nested case-control (from a cohort study) | T = 200 (Ct = 100; depressed = 100) | Caucasian: 81% in depressed group and 78% in Cts | NR | DSM-V diagnosis of depression (depressed group). Documented negative EPDS for Cts | Prior to pregnancy and confirmed during pregnancy | Serum/NR | 11–14 gw (M = 12.8 gw in Cts and 12.6 gw in depressed group) | (1) hs-ELISA; (2) Multiplex Assay | (1) IL-6, (2) TNF-α | ↑ IL-6 (p =.03) and ↑ TNF-α (p < .0001) levels in depressed vs Cts groups. | 6 |
| [44] | Thailand | Cross sectional | T = 73 (Non-pregnant = 24; pregnant no PND = 25, pregnant + PND = 24) | Years of education: 17.8 ±1.3 Non-pregnant; 12.5 ±3.3 pregnant no PND; 12.0 ±4.3 pregnant + PND. | NR | EPDS to asses the diagnosis of perinatal depression (EPDS≥11 for depressed and EPDS≤2 for non-depressed) and HAM-D | 3rd trimester (at the end of term) and 4–6 wks PP | Serum | 3rd trimester (at the end of term) | (1) ELISA; (2) Atomic absorption spectrometric method; (3) latex immunoassay; (4) spectrophotometric method; | (1) assay of IgM/IgA responses to tryptophan + TRYCATS: Anthranilic acid, Xanthurenic acid, Quinolinic acid, Kynurenic acid, IgA Picolinic acid, 3-OH Kynurenine, Kynurenine, Quinaldic acid, 3-OH Anthranilic acid, (2) zinc, (3) hs-CRP, and (4) haptoglobin | ↓ IgA responses to anthranilic acid in PND vs pregnant no PND (p =.047). | 8 |

*(Continued)*

**Table 1.** (Continued)

| Reference | Country | Study design | Number of subjects | Socio-economic status/ethnicity | Delivery mode | Assessment of depression | | Biological fluid/hour of collection | Inflammatory protein markers | | | Results | Total quality score |
|---|---|---|---|---|---|---|---|---|---|---|---|---|---|
| | | | | | | Instruments | Timepoint(s) | | Timepoint(s) | Dosage assessment technique | Inflammatory markers | | |
| [45] | Sweden | Case-control (substudy in the BASIC cohort study) | T = 258 (Ct = 160; Perinatal depressive disorder = 59; Antidepressant treatment = 39) | University education: Ct = 84.4%, perinatal depressive disorder = 67.8%, antidepressant treatment = 74.4% | NR | (1) EPDS (psychophysiology sub-study EPDS≥13, cesarean EPDS≥17). (2) MINI and MADRS performed only in participants from the psychophysiology sub-study. | 17 and 32 gw; (2) 35–39 gw | Plasma/9:30am–1:00pm and morning before cesarean | Late pregnancy (35–39 gw) (n = 205) and at morning before cesarean section (38w) (n = 53) | Proseek Multiplex Inflammation I (based on proximity extension assay technology) | 92 inflammatory proteins | ↓23 inflammatory proteins in women with perinatal depression (p < .01) and women on SSRI treatment in comparison with Ct (p < .01): TRAIL, CSF1, CX3CL1, CST5, DNER, VEGFA, STAMPB, CD5, CD244, TNFRSF9, TNFB, IL10RB, CD40, IL15RA, hGDNF, ST1A1, CCL11, ADA, CCL25, uPA, AXIN1, SLAMF1, IL17C (-) significant correlations between the 23 molecules described above with depression symptoms severity (MADRS scores) at 36 gw as well as with EPDS scores in 17 and 32 gw (with the exception of AXIN1 and TNFRSF9 at 32 gw). | 6 |
| [46] | USA | Cross-sectional (from a prospective observational study) | T = 117 | NR | 100% cesarean | IDS-SR30≥18, MINI | Pre-cesarean | Plasma and CSF/NR | Pre-cesarean | Flow cytometry | IL-1β, IFN-α, IFN-γ, TNF-α, MCP-1, IL-6, IL-8, IL-10, IL-12p70, IL-17A, IL-18, IL-23, and IL-33. | ↑ IL-1β (95%CI 5.9–9148.5), ↑ IL-23 (95%CI, 1.7–294.5), ↑ IL-33 (95%CI 1.1–2.6) levels in the CSF have significant associations with increased odds of PND. | - |
| [47] | China (Taiwan) | Case-control | T = 33 (Ct = 16; PND = 17) | Education years: Ct = 16.8±1.0, PND = 16.6±1.5 | NR | DSM-IV diagnosis by MINI; EPDS≥12/13 | 16–28 gw | Plasma/9:00–10:00am | gw | (1) nephelometry immunoassay; (2) Multiplex assay | (1) CRP; (2) IL-6, IL-10 and TNF-α | ↑TNF-α levels in PND vs Ct groups (p = .016). (+) correlation between PND duration and TNF-α levels (p < .01) | 6 |

*(Continued)*

Table 1. (Continued)

| Reference | Country | Study design | Number of subjects | Socio-economic status/ethnicity | Delivery mode | Assessment of depression | | Inflammatory protein markers | | | | Results | Total quality score |
|---|---|---|---|---|---|---|---|---|---|---|---|---|---|
| | | | | | | Instruments | Timepoint(s) | Biological fluid/hour of collection | Timepoint(s) | Dosage assessment technique | Inflammatory markers | | |
| [48] | USA | Cross-sectional | T = 105 | 51.5% High school or less; 4.8% household income <$4,999; 70.5% caucasian | NR | POMS-D continuous total sum score (7,6% met the protocol's screening criteria for possible clinical depression with a POMS-D score greater than 20) | 16–26 gw (M = 20 gw) | Plasma/mean time of day = 11:25am | 16–26 gw (M = 20 gw) | 12-plex assay | TNF-α, IFN-γ, IL-1β, IL-2, IL-4,IL-5, IL-6, IL-7, IL-8, IL-10, IL-12, and IL-13 | (-) correlations of POMS-D with IL-1β (p = < .05), IL-7 (p < .05) and TNF-α (p < .01). Hierarchical linear regression models: depression contributed to the levels of plasma cytokine IL-1β, IL-7 and TNF-α. | 6 |
| [49] | USA | Cross-sectional (from a prospective observational study) | T = 206 | 95.1% High school or less; 49.5% Mexican American | NR | Depressive symptoms defined as a CES-D score of >20. | 22–24 gw | Plasma/NR | 22–24 gw | ELISA | IL-1RA | ↑IL-1RA levels in group with CES-D>20 in comparison with CES-D<20 group (p = .018). | 8 |
| [50] | USA | Cross-sectional (from a cohort study) | T = 72 | African-american; 60% unemployed; 49% high school level or less | NR | CES-D continuous total sum score (CES-D ≥ 16 frequently used to indicate a positive screen for depression) | 14–17 gw | Plasma/NR | 14–17 gw | BioPlex Pro 17-plex kit. | IL-1β, TNF-α, IL-6, IL-8, IL-12, IL-17, IFN-γ | (+) correlations between CES-D scores and IL-8 (p = .008). | 6 |
| [51] | NR | Cross-sectional | T = 44 (Ct = 14; PMMD = 30) | NR | NR | PMMD monitored by consultation psychiatric service. The severity of depressive symptoms was measured by the EPDS. | M = 32.9 ± 4.18 gw | Plasma/ Late Morning | M = 32.9; SD = 4.18 gw | ELISA kit | ERVWE1 | NS | 7 |
| [52] | Italy | Cross-sectional study | T = 79 women (20 with PND + TRAUMA, ii) 19 with PND + no-TRAUMA, iii) 20 HV pregnant; iv) 20 HV non-pregnant | Years of Education: No trauma = 13.0±3.3; Trauma = 14.5±2.9; HVpregnancy = 14.0 ±3.9); Unemployed: No trauma = 10.5%; Trauma = 25%; HVpregnancy = 68.8% | NR | Diagnosis of PND was made by clinical interview according to the DSM-V criteria. The severity of current depressive symptoms was assessed by EPDS (EPDS≥12 for clinically relevant PND) | 22–24 gw | Serum/8:00 and 10:00 a.m | 22–24 gw | (1) NR; (2) ELISA | 1) CRP; 2) TNF-α and IL-6 | ↑ CRP in the PND + no-TRAUMA group compared to the HV pregnant group; | 6 |

(*Continued*)

**Postpartum**

Table 1. (Continued)

| Reference | Country | Study design | Number of subjects | Socio-economic status/ethnicity | Delivery mode | Assessment of depression | | Inflammatory protein markers | | | | Results | Total quality score |
|---|---|---|---|---|---|---|---|---|---|---|---|---|---|
| | | | | | | Instruments | Timepoint(s) | Biological fluid/hour of collection | Timepoint(s) | Dosage assessment technique | Inflammatory markers | | |
| [53] | USA | Cross-sectional (from a longitudinal study) | T = 69 | Annual household income <$15,000: 37,5% African American group and 18.9% in White group. High school graduate or less: 15.6% African American group and 27.0% in White group. African American (n = 32) and White (n = 37) | NR | CES-D continuous total sum score | 7–10 wks PP | Serum and LPS-stimulated PBMCs culture/NR | 7–10 wks PP | (1) single spot ultra-sensitive ELISA; (2) multiplex assay | Serum: (1) IL-6; (2) TNF-α and IL-8 LPS-stimulated PBMCs: (2) IL-6, TNF-α, IL-8 and IL-1β | Among African American women, controlling for BMI, (+) correlation between stimulated IL-6 (p≤.01), IL-8 (p≤.01) and TNF-α (p = .02) and CES-D depressed mood subscale. | 8 |
| [54] | USA | Cross-sectional (from a RCT study) | T = 56 (n = 33 for hsIL-6 ELISA) | 23.2% lower SE class; 92.9% Caucasian | NR | (1) HRSD≥15 and (2) clinical interview including SCID-IV (if HRSD≥15 twice within the following 7-day period to identify a depression recurrence). | (1) 2,3,4,6,8,11,14 and 17 wks PP; | Plasma/NR | 2, 3, or 4 wks PP | hsELISA | hsIL-6 | NS (for risk depression recurrence) | 7 |
| [55] | China | Cross-sectional | T = 10 (Ct = 8; PPD = 2) | NR | NR | patients with PPD provided by medical patterns | within the 4 weeks period of giving birth | Plasma/NR | within the 4 weeks period of giving birth | Ultrasensitive Lab-on-chips (LOC) device and ELISA | IL-2, IL-6, IL-8 and TNF-α | ↑IL-2 and ↑IL-8 levels in PPD compared to Ct group (no statistical information is reported) | 2 |
| [56] | USA | Cross-sectional | T = 181 | Income <$10,000: 13.6% breastfeeders group and 39.4% formula feeders group. White race: 89.8% breastfeeders group and 73.4% formula feeders group. | NR | POMS-D continuous total sum score | between 4–6 wks PP (M = 5.2 wks) | Serum/8:00–11:00 am | between 4–6 wks PP (M = 5.2 wks) | ELISA | IFN-γ and IL-10 | (-) correlation between POMS-D and INF-γ only in the formula feeders (p < .05). | 8 |
| [57] | USA | Cross-sectional (secondary analysis) | T = 199 (depressed n = 25) | Depressed mothers had lower income in comparison with no-depressed women (p < .06) | NR | POMS-D ≥21 (depressed mothers were categorized as those with scores in the highest decile on the POMS-D scale) | 4–6 wks PP (M = 5.3 wks PP) | Serum and ex-vivo whole blood culture supernatant/ 8:00–11:00 am | 4–6 wks PP (M = 5.3 wks PP) | ELISA | IFN-γ, IL-6 IL-10 | ↓ serum IFN-γ (p < .001), ↓IFN-γ/ IL-10 ratio in both serum (p < .04) and whole blood culture supernatants (p < .009) in depressed in comparison with not-depressed. | 6 |
| [58] | Iran | Case-control | T = 62 (Cts = 30; MDD = 10; minor depression = 22) | NR | NR | a psychiatrist used DSM-IV-TR to diagnose major and minor depression. | NR | Serum/NR | immediately after parturition | single radial immunodiffusion | IgG, IgM and IgA and complements C3 and C4 | ↑IgG levels in MDD (p = .026) compared with minor depression | 6 |

(Continued)

**Table 1.** (Continued)

| Reference | Country | Study design | Number of subjects | Socio-economic status/ethnicity | Delivery mode | Assessment of depression | | Biological fluid/hour of collection | Inflammatory protein markers | | | Results | Total quality score |
|---|---|---|---|---|---|---|---|---|---|---|---|---|---|
| | | | | | | Instruments | Timepoint(s) | | Timepoint(s) | Dosage assessment technique | Inflammatory markers | | |
| [59] | USA | Cross-sectional | T = 165 (Cts = 60; depressive episode with peripartum onset = 87; 18 displayed depression but did not fulfill the time criteria for peripartum-onset depression defined by DSM-5). | Unemployed: Cts = 38%; Depressive episode with peripartum onset = 37% | NR | EPDS total score; clinical interview SCID-5 for diagnosis of depressive episode with peripartum onset | 8 wks PP (6–12 wks PP) | Plasma/9:00 am–12:00 pm | 8 wks PP (6–12 wks PP) | (1) MesoScale Discovery platform; (2) HPLC; (3) GC-MS; (4) UPLC-MS/MS | (1) IL-1β, IL-2, IL-6, IL-8, IL-10, and TNF-α; (2) Tryptophan, Kynurenine; (3) quinolinic acid; (4) serotonin, kynurenic acid and nicotinic acid | ↑IL-6, ↑IL-8 and ↓serotonin, ↓IL-2 and ↓quinolinic acid were associated with ↑ risk of PPD ($OR_{IL-6}$ = 3.0, p = .007; $OR_{IL-8}$ = 3.32, p = .009, per pg/ml increase; $OR_{serotonin}$ = 1.43, p = .016, per nM decrease; $OR_{IL-2}$ = 2.34, p = .002, per pg/ml decrease; $OR_{quinolinic\ acid}$ = 4.48, p = .014, per nM decrease) and ↑ depressive symptoms (IL-6: p = .022; IL-8: p = .006; serotonin: p = .003; IL-2: p = .005; quinolinic acid: p = .022) ↑kynurenine/ serotonin ratio was associated with an increased risk for PPD (OR = 1.35 per unit increase, p = .038) and ↓serotonin/ kynurenine ratio was associated with ↑EPDS score (p = .009). | 7 |
| [60] | Japan | Cross-sectional | T = 129 breastfeeding mothers | 96.1% high social support; 9 or less years of education: 1.6% | NR | EPDS ≥ 9 as having postpartum depression. | 3 mo PP | Breast milk/ 65.9% at 12:00–2:00 pm | 3 mo PP | ELISA | TGF-β2 | ↑TGF-β2 in mothers with depression than mothers without depression (p = .02); depressed mothers had a 3.11-fold (95% CI: 1.03–9.37) higher likelihood to have ↑TGF-β2 levels in their breast milk. | 8 |
| [61] | Sweden | Cross-sectional | T = 64 (preterm group = 27; term group = 37) | Low annual income (<115,000 SEK): preterm = 0%, term = 10.8%. Secondary school: preterm = 51.8%, term = 40.5%. | NR | Semi-structured interview. Descriptions of continuous depressed mood were considered as depressive symptoms. | Within 5 days after delivery | Serum/NR | During labor | Multiplex assay | IL-1α, IL-1β, IL-2, IL-4, IL-6, IL-8, IL-10, IL-12 p70, IL-13, IL-17, IL-18 and IFN-γ | Preterm group: (+) correlation between IL-8 and depressive symptoms (p < .01) | 6 |

(Continued)

**Table 1.** (Continued)

| Reference | Country | Study design | Number of subjects | Socio-economic status/ ethnicity | Delivery mode | Assessment of depression | | | Inflammatory protein markers | | | Results | Total quality score |
|---|---|---|---|---|---|---|---|---|---|---|---|---|---|
| | | | | | | Instruments | Timepoint(s) | Biological fluid/hour of collection | Timepoint(s) | Dosage assessment technique | Inflammatory markers | | |
| [62] | USA | Cross-sectional | T = 119 | Income mean $18,000 ± $6,000. 84% Caucasian | NR | POMS-D total score (cutoff score of 25 above which triggered a referral to a mental health professional.) | 4–6 wks PP (M = 4.5 ± 2.3) | Morning hindmilk/pick up visit before 11:30 am | 4–6 wks PP (M = 4.5 ± 2.3) | ELISA | IgA | NS | 9 |
| [63] | USA | Cross-sectional (from multicenter urban birth cohort study) | T = 469 | 40% less than high school education. 70% annual income <$15,000. 73% African American | NR | EPDS total score (EPDS≥12 indicated a need for further mental health evaluation) | 89% at 12 mo PP, 8% at 24 mo and 3% at 36 mo PP | PBMC cells culture ex-vivo estimulated/ NR | 89% at 12 mo PP, 8% at 24 mo PP and 3% at 36 mo PP | Multiplex ELISA assay | Innate Stimuli: IFN-α, IFN-γ, IL-10, IL-12p40, TNF-α and IL-8. Adaptive and Mitogenic Stimuli: IFN-γ, IL-10, IL-13, IL-4 and IL-5 | (-) correlations between depression and innate immune responses, namely, CpG-induced TNF-α (p = .02), RSV induced IL-8 (p < .05), and LPS induced IFN-γ (p = .02). (-) correlations between depression and adaptive immune responses, namely, several DM-induced cytokines (IL-4, IL-5, IL-10, and IL-13; p < .05), and CR-induced IL-10 (p = .01). | 7 |

Note: ADA: Adenosine deaminase; AXIN1: Axin 1; BMI: Body Mass Index; CES-D: Center for Epidemiological Studies Depression Scale; CCL: C-C Motif Chemokine Ligand; CpG: Cytosine-phosphate-guanine; CR: Cockroach extract; CRP: C-reactive protein; CSF: Cerebrospinal fluid; CSF1: Colony stimulating factor 1; CST5: Cystatin D; Cts: Controls; CX3CL1: C-X3-C Motif Chemokine Ligand 1; DM: Dust mite* extract (* D. pteronyssinus); DNER: Delta/Notch Like EGF Repeat Containing; DSM: Diagnostic and Statistical Manual of Mental Disorders; ELISA: Enzyme-linked immunosorbent assay; EPDS: Edinburgh Postnatal Depression Scale; ERVWE1: Human Endogenous Retrovirus W EnvC7-1 Envelope Protein; GC-MC: Gas chromatography-mass spectrometry; gw: Gestational weeks; HAM-D or HDRS: Hamilton Depression Rating Scale; hGDNF: Glial cell line-derived neurotrophic factor; HPLC: High performance liquid chromatography; hs: High-sensitivity; HV: Healthy volunteers; IDS-SR30: Inventory of Depressive Symptomatology Self-Rated; IFN-α: Interferon alpha; IFN-γ: Interferon gamma; Ig: Immunoglobulins; IL-: Interleukin-; IL-1β: Interleukin 1 beta; IL-1RA: Interleukin 1 receptor antagonist; IL10RB: Interleukin 10 Receptor Subunit Beta; IL15RA: Interleukin 15 Receptor Subunit Alpha; LPS: Lipopolysaccharides; M: Mean; MADRS: Montgomery Asberg Depression Rating Scale; MCP-1: Monocyte chemotactic protein 1; MDD: Major Depressive Disorder; MIF: Macrophage migration inhibitory factor; MINI: The Mini International Neuropsychiatric Interview; mo: Months; ng/ml: Nanogram per milliliter; nM: Nanomolar; NR: Not reported; NS: Not significant results; PBMCs: Human peripheral blood mononuclear cells; PMMD: Perinatal Major Depressive Disorder; PND: Perinatal depression; POMS-D: Profile of Mood States-depression–dejection scale; PP: Postpartum; PPD: Postpartum depression; RCT: Randomized Control Trial; RSV: Respiratory syncytial virus; SCI: Structured clinical interview; SCID: Structured Clinical Interview for DSM; SE: Socio-economic; SLAMF1: Signaling Lymphocytic Activation Molecule Family Member 1; STAMPB: STAM Binding Protein; ST1A1: Sulfotransferase Family 1A Member 1u; T: Total; TGF-β2: Transforming growth factor-beta 2; TNF-α: Tumor necrosis factor alfa; TNFB: Tumor necrosis factor B; TNFRSF9: TNF Receptor Superfamily Member 9; TRAIL: TNF-related apoptosis-inducing ligand; uPA: Urokinase-type plasminogen activator; UPLC-MS/MS: Ultra-high performance liquid chromatography, coupled to tandem mass spectrometry; VEGFA: Vascular Endothelial Growth Factor A; vs: Versus; wks: Weeks; ↑: Increase; ↓: Decrease; (+): Positive; (-): Negative.

IL-6, IL-10, and TNF-α were chosen as they are the most relevant cytokines studied in the field of depression research and represent both proinflammatory (IL-6 and TNF-α) and anti-inflammatory cytokines (IL-10).

**Table 2. Longitudinal studies of inflammatory biomarkers and depression.**

| Reference | Country | Study design | Number of subjects | Socio-economic status/ethnicity | Delivery mode | Assessment of depression | | Biological fluid/hour of collection | Inflammatory protein markers | | | Results | Total quality score |
|---|---|---|---|---|---|---|---|---|---|---|---|---|---|
| | | | | | | Instruments | Timepoint(s) | | Timepoint(s) | Dosage assessment technique | Inflammatory markers | | |
| **Pregnancy** | | | | | | | | | | | | | |
| [64] | Canada | Longitudinal | T = 27 | 48% had some high school education; 65% annual family income <$40,000. | NR | PHQ-9 continuous scores to represent symptom severity. | T1 (7–10 gw), T2 (16–20 gw) | Serum/08:30 am–12:30am | T1 (7–10 gw), T2 (16–20 gw) | ELISA | CRP, hsIL-6 and hsTNF-α | The inflammatory markers at T1 and depressive symptoms at T1 and T2 were moderately correlated (r = 0.44–0.53). ↑ depressive symptoms at T1 predict ↑CRP levels (p = .03) and ↑TNF-a levels (p = .00) at T2. ↑ depressive symptoms at T2 predict ↑CRP levels (p = .01), ↑IL-6 levels (p = .04) and ↑TNF-a levels (p = .047) at T2. There was a significant association between increase in depressive symptoms from T1 to T2 and IL-6 levels (p < .006). | 5 |
| [65] | USA | Longitudinal (from a prospective cohort study) | T = 145 | 20.7% incomplete high school; 46.2% African American | 19% cesarean | EPDS continuous total score, SCID (Clinical diagnoses of current depression and history of depressive episodes) | 18 and 32 gw | Serum/ 8:00am–14:00pm; 14 samples were collected between 14:00–16:00pm | 18 and 32 gw | hsELISA | IL-6 and TNF-α | NS | 6 |
| [66] | Finland | Longitudinal (from a Cohort Study PREDO) | T = 295 (sample with depressive symptoms reported during pregnancy) | NR | NR | CES-D (continuous variable and as a binary variable indicating probable clinical depression-CES-D≥16). | 2x wk until 38–39 gw or delivery. | Plasma/ 19:00 PM–21:00 PM | median 13, 19, 27 gw | CRP immunoturbidimetric assay | hsCRP | ↑hsCRP levels in those with compared to those without depressive symptoms during pregnancy (95% CI 0.17–1.88 mg/L). | 8 |
| [67] | Ireland | Cohort study | T = 209 healthy pregnant women (n = 104 with no IBS) | NR | NR | EPDS (Lower quartile (PSS—7.5, STAI—23.3; EPDS—2.5) and upper quartile scores (PSS—17; STAI—40; EPDS—9.5) were used as cutoffs to define low- (<25th percentile), moderate- (25th to <75th percentile), and high- (≥75th percentile) scoring groups for each of the psychological evaluations. | 15 ± 1 (visit 1) and 20 ± 1 (visit 2) gws | Plasma + Serum / before 12 pm on the morning of each visit | 15 ± 1 (visit 1) and 20 ± 1 (visit 2) wks' gestation | (1) Multiplex Assay; (2) HPLC | 1) IFN-γ, TNF-α, IL-6, IL-18, IL-8, IL-10, MCP-1, SDF-1α, MIF and CRP; 2) TRP and KYN | ↑CRP levels in the high-scoring group vs moderate (p = .026) and low scores (p = .048). ↑TNF-α in the moderate-scoring group vs the low-scoring group (p = .006). ↓IL-8 levels at 20 gw vs at 15 gw for moderate- (p = .001) and high-scoring groups (p = .035). (-) correlation between IL-8 levels and EPDS scores at 20 gw (NS in the adjusted model) | 7 |
| **Pregnancy and Postpartum** | | | | | | | | | | | | | |
| [68] | USA | Longitudinal | 152 women | 80% married 76% Caucasian 24.4% low income | 100% vaginal delivery | Symptomatic of PPD (EPDS≥10) | 32–36 gw, 7d, 14d ± 48-h, 1,2,3,6 mo (± 1w) PP | Plasma/NR | 32–36 gw, 7d, 14d ± 48-h, 1,2,3,6 mo (± 1w) PP | Human Pro-inflammatory Ultra-Sensitive assay and quantitative multiplex array technology (not discriminated) | IL-6, IL-1β, TNF-α, IL-8, IFN-γ, IL-10. | ↓TNF-α in PPD vs not depressed at any time (p < .05) (results replicated over time) ↑ day 14 IL-8/IL-10 ratio among PPD symptomatic vs. non-symptomatic (p = .006). | 6 |

*(Continued)*

**Table 2.** (Continued)

| Reference | Country | Study design | Number of subjects | Socio-economic status/ethnicity | Delivery mode | Assessment of depression — Instruments | Assessment of depression — Timepoint(s) | Biological fluid/hour of collection | Inflammatory protein markers — Timepoint(s) | Dosage assessment technique | Inflammatory markers | Results | Total quality score |
|---|---|---|---|---|---|---|---|---|---|---|---|---|---|
| [69]* | Japan | Longitudinal | 132 pregnant women (ND = 62, PD = 15, TG = 22; CD = 33) | NR | NR | EPDS (ND group = EPDS<8/9 all timepoints; PD group = EPDS>8/9 only at the 1 mo PP period; TG group = EPDS>8/9 only during pregnancy; CD group = EDPS>8/9 during pregnancy and PP. | before 25 gw and ~36 gw; 1mo PP | Plasma /NR | before and 1 mo after delivery | HPLC | TRP, KYN, KA, AA, 3HK and 3HAA. | ↑ KYN and ↑ KA (p < .01) and ↑ KYN/TRP and ↑ KA/KYN ratio (p < .05) in PD group vs ND group during pregnancy period. ↓ 3HAA PD vs ND group in PP period (p < .05). ↓ KYN ratio (p < .01) and ↓ KYN/TRP ratio (p < .01) (during PP period to that during pregnancy) in PD group vs ND group. (+) correlations between KYN (p < .01), KA (p < .05) levels and KYN/TRP ratio (p < .05) during pregnancy and EPDS score during postnatal period. (-) correlations between 3HAA levels (p < .05) and EPDS score during postnatal period. | 5 |
| [70] | USA | Longitudinal cohort study | 171 pregnant women (81 African American group, 90 non -African American group ) | 53% single | 34% cesarean section | EPDS; SCID—current diagnoses of depression | 18 gw, 32 gw, 6 w PP, 6 mo PP | Serum | 18 gw, 32 gw, 6 wks, 6 mo PP | hs ELISA | IL-6, TNF-α | NS | 5 |
| [71] | Belgium | Longitudinal | 98 healthy pregnant females | NR | NR | (1) BDI, (2) SCID diagnoses of PPD | (1) 3 –6 days before the anticipated delivery + 1 and 3 days PP; (2) 6–10 m PP | (1) Plasma; (2) Serum | 08:00 am (+/-30 min) 3–6 days before the anticipated date of delivery and 1 and 3 days after delivery | (1) HPLC, (2) ELISA | (1) TRP and KYN; (2) IL-6, IL-8, LIF-R | ↑ KYN (p = .01) and ↑ K/T quotient (p = .01) in BDI responders vs. nonresponders 3d after delivery ↑ in the K/T quotient over time in BDI responders (p = .0005) vs. NS changes overtime in non-responders. | 7 |
| [72] | Belgium | Longitudinal | 71 participants, (15.5% major PPD, 8.4% minor PPD) | NR | NR | (1) ZDS; (2) SCID diagnoses of PP depression | (1) 3–5 days before the anticipated date of delivery and 1 and 3 days after delivery; (2) 6–10 m PP | Serum | 08:00 am (+/-30 min) 3–5 days before the anticipated date of delivery and 1 and 3 days after delivery | ELISA | CC16 | ↓ serum CC16 levels (p = .002) in women with PPD vs those without postpartum depression. | 8 |
| [73] | Belgium | Longitudinal | 98 healthy pregnant females | NR | NR | (1) ZDS; (2) SCID diagnoses of PP depression | (1) 3–5 days before the anticipated date of delivery and 1 and 3 days after delivery; (2) 6–10 m PP | Plasma | 08:00 am (+/-30 min) 3–5 days before the anticipated date of delivery and 1 and 3 days after delivery | HPLC | Tryptophan | NS | 7 |
| [74]* | USA | Longitudinal (secondary analysis) | T = 63 | 19.4% high school or less, 64.5% White race | NR | EPDS≥12 | At baseline (between 28–34 gws M = 32.5 gws), 3 mo PP, 6 mo PP | Serum/8:00 am–5:00 pm | At baseline (between 28–34 gws M = 32.5 gws), 3 mo PP, 6 mo PP | Multiplex assay | TNF-α, IL-6, IL-1β and CRP | ↑TNF-α associated with ↓EPDS score (95% CI [−1.84,−.036]), adjusted for confounders (days since delivery, time of blood draw and antidepressant treatment or potentially anti-inflammatory medications and self-reported level of hardship) | 6 |

*(Continued)*

**Table 2.** (Continued)

| Reference | Country | Study design | Number of subjects | Socio-economic status/ ethnicity | Delivery mode | Assessment of depression | | Inflammatory protein markers | | | | Results | Total quality score |
|---|---|---|---|---|---|---|---|---|---|---|---|---|---|
| | | | | | | Instruments | Timepoint(s) | Biological fluid/hour of collection | Timepoint(s) | Dosage assessment technique | Inflammatory markers | | |
| [75] | USA | Longitudinal | 151 pregnant women | 78.1% Caucasian 24.5% low income | 100% vaginal | EPDS | 32–36 gw, 1 w PP, 2 w PP, 1 mo PP, 2 mo PP, 3 mo PP and 6 mo PP (7 total) | Blood | 32–36 gw, 1 w PP, 2 w PP, 1 mo PP, 2 mo PP, 3 mo PP and 6 mo PP (7 total) | NR | IL-6, TNF-α, IL-10 | ↑TNF-α associated with ↓EPDS (95%CI = [−1.24,−0.11]) | 6 |
| [76] | Belgium | Longitudinal | 91 healthy pegnant women. | NR | 100% vaginal | (1) ZDS; (2) SCID diagnoses of PP depression | 3–5 days before the anticipated delivery, 1 day PP and 3 days PP; (2) 6–10 m PP | serum | 3–5 days before the anticipated delivery, 1 day PP and 3 days PP | ELISA | IL-6, IL-6R, sgp130, IL-1RA and LIFR | ↑IL-6 (p = .027) and ↑ IL-6R (p < .001) in ZDS reactors vs. non-reactors, before delivery (p = .006), and 1d (p = .001) and 3d (p = .002) PP; ↑IL-6 ↑IL-6R values in reactors vs. non-reactors (p < .001), 1d (p = .001) and 3d (p = .01) after delivery | 6 |
| [77] | USA | Longitudinal | 28 pregnant women (14 healthy controls, 14 depressed group = EPDS ≥ 10) | Less than $20,000 14.3% (healthy) and 28.6% (depressed); 50% Caucasian and 42.9% African American in both groups; less than high school 7.1% (healthy) and 21.4% (depressed) | Vaginal: 85.7% Healthy, and 70% Depressed | EPDS | 8–12 gw | Plasma | 8–12 gw (visit 1),24–28 gw (visit 2), and 6–8 w PP (visit 3) | (1) limulus amebocyte lysate QCL-1000 Kit; (2) Bio-Plex Pro™Human Chemokine Panel | (1) LPS; (2) TNF-α, IL-6, IL-1β and MCP/CCL2 | ↑ LPS at 8–12 gw (visit1) (p = .007), ↑ TNF-α and MCP/CCL2 at 8–12 gw (p = .02 and p = .04, respectively) and ↑IL-6 at 24–28 gw (visit2) (p = .02) in depressive subjects vs. healthy group, even when BMI, age and race were controlled ; NS differences between depressed vs. healthy women for IL-6 and IL-1β at visit 1 | 8 |
| [78]* | Sweden | Nested case-control (from BASIC cohort study) | T = 291 (Cts = 228; PP depressive symptoms = 63) | Primary/Secondary School: Cts = 19.7% and PP depressive symptoms = 30.2% | Vaginal or Vacuum Extraction: Cts = 61.4%, PP depressive symptoms = 52.4% | EPDS≥14 and MINI interview | 8 wks PP | Plasma/most at morning | Days from blood sampling to delivery: Cts = 14.0 (median), 15.0 (IQR); | Multiplex extension assay | 74 of 92 inflammatory markers (16 excluded for being below LOD for >50% of the samples and 2 for technical problems) | ↑STAM-BP (p = .002), ↑Axin-1 (p = .004), ↑ADA (p = .001), ↑ST1A1 (p = .04) and ↑IL-10 (p = .029) NPX values in Cts vs depressive symptoms groups after adjusting for multiple testing. ↑STAM-BP (p = .07), ↑Axin1 (p = .007), ↑ADA (p = .020), ↑ST1A1 (p = .026), ↑SIRT2 (p = .016), ↑CASP8 (p = .013), ↑IL-10 (p = .039) and ↑MCP2 (p = .015) NPX values in Cts vs PP depressive symptoms group. | 6 |
| [79] | USA | Longitudinal | 51 women | black: more depressed (58%) ,less depressed (42%), hispanic: more depressed (42%), less depressed (53%), partner: more depressed (58%), less depressed (72%) | Pre-term birth: more depressed (17%), less depressed (16%) | BDI scores (≤ 9 vs. >9) | T1: 8–20 gw (M = 14.5, SD = 3.1; T2: 26 gw (SD+1); T3 = 35 gw (SD = 0.8); T4: 6wks pp; T5: 24 wks pp | Serum | T1: 8–20 gw (M = 14.5, SD = 3.1; T2: 26 gw (SD+1); T3 = 35 gw (SD = 0.8); T4: 6wks pp; T5: 24 wks pp | Bead-based ELISA | 23 cytokines | (↑) IL-6 and CCL3 at T3 (IL-6p < .001; CCL3: p < .001), (↑) IL-15 at T1 and T3 (p = .027 and p = .003, respectively), (↑) G-CSF at (T4) PP1 (p = .002) on more depressed vs. less depressed woman Slope of change for cytokines (IL-6, CCL3, IL-15, G-CSF) are different between depressed and non-depressed women. | 7 |

(Continued)

**Table 2.** (Continued)

| Reference | Country | Study design | Number of subjects | Socio-economic status/ethnicity | Delivery mode | Assessment of depression | | Inflammatory protein markers | | | | Results | Total quality score |
|---|---|---|---|---|---|---|---|---|---|---|---|---|---|
| | | | | | | Instruments | Timepoint(s) | Biological fluid/hour of collection | Timepoint(s) | Dosage assessment technique | Inflammatory markers | | |
| [80] | Thailand | Longitudinal | 24 non pregnant women, 25 non-depression pregnant women, 23 pregnant women with depression | education in years: non pregnant = 7.8 (1.3), pregnant no depression = 12.5(3.3), pregnant depression = 11.9 (4.3); Single/married/separated: non pregnant = 21/3/0, pregnant no depression = 4/18/3, pregnant depression = 4/17/2 | NR | EPDS ≥11 categorized as "prenatal depression", BDI, HAM-D, MINI-Thai version | 3rd pregnancy trimester and 4–6 w PP | Plasma | Third trimester and 4–6 w PP | hs CRP Vario assay | hs-CRP | (+) correlation between CRP and prenatal EPDS (p = .007), HAM-D (p = .013), and BDI (p = .001) | 6 |
| [81] overlap sample with [82] | USA | Longitudinal | 29 pregnant women | African American (85%) and single (74%) | NR | SIGH-SAD | T1(35–38 gw), T2 (1–5 d PP), T3 (5–6 wks PP) | Serum | T1(35–38 gw), T2 (1–5 d PP), T3 (5–6 wks PP) | (1) ELISA (2) HPLC on reversed phase | (1) CRP, IL-6, (2) TRP, KYN and KYN/TRP | (+) correlation between prepartum CRP and SIGH-SAD atypical score (p = .007) (-) correlation between early pp CRP and SIGH-SAD atypical score (p = .047) | 6 |
| [83] | USA | Longitudinal | 46 pregnant women (only 12 have completed surveys and blood samples at both time points) | 58.3% African American, 66.7% high school or higher education, 16.7% married or partnered, 66.7% were single never married, 25% were full-time employed and 3.3% were part-time employed | 50% C-section | CES-D | T1 (>36 gw); T2 (4–6 wks PP) | Serum | T1 (>36 gw); T2 (4–6 wks PP) | Bioplex Cytokine Assay | L-1β, IL-2, IL-4, IL-5, IL-6, IL-7, IL-8, IL-10, IL-12, IL-13, IL-17, MIP-1α, GM-CSF, IFN-γ, MCP-1, MIP-1β, and TNF-α | (+) correlation between prenatal depressive symptoms and MIP-1β (p < .05) | 4 |
| [84] | Germany | longitudinal | 100 pregnant woman | Married or solid partnership (98.6%); | NR | MARDS and EPDS | 34 gw, 38 gw, 3 days PP, 7 wks PP, 6 mo PP | Serum/8-10 am | 34 gw, 38 gw, 3 days PP, 7 wks PP, 6 mo PP | ELISA | neopterin | ↑ postnatal neopterin levels in mothers with postnatal depressive symptoms vs. mothers without postnatal depressive symptoms (p = .049) using MARDS. | 7 |
| [82] overlap sample with [81] | USA | longitudinal | 27 women | African American (85%), never married (74%), education: less than high school (19%), high school graduate (48%) | NR | SIGH-SAD | 35–38 gw; 1–5 days PP, 5–6 wks PP | Serum/NR | 35–38 gw; 1–5 days PP, 5–6 wks PP | (1) Two antibody ELISAs (2) HPLC on reversed phase | (1) CRP, IL-6, (2) TRP, KYN and KYN/TRP | (+) correlations between CRP and prepartum atypical depression scores (p = .005) as well in total depression score. (-) correlations between CRP total and atypical depression scores in the early PP. (-) correlations between TRP and total depression score in the prepartum period. | 6 |
| [85]* | Canada | Longitudinal | 33 healthy pregnant women | working full-time (80.7%); with university degree (51.7%); Married/common (100%) | NR | EPDS | 3rd trimester (M = 30.1, SD = 4.1) and 12 w PP (M = 13.5, SD = 1.9) | Serum/8.15 am—3:30 pm | 3rd trimester (M = 30.1, SD = 4.1) and 12 w PP (M = 13.5, SD = 1.9) | ELISA | IL-6, IL-10, TNF-α, and CRP | ↓ IL-6 (p = .025), IL-10 (p = .006) at 3rd trim predict ↑ PP EPDS scores | 7 |
| [86]* | Sweden | Case-control | 347 women | NR | NR | EPDS | 5 days PP, 6 wks PP and 6 mo PP | Serum/NR | Before delivery | ELISA | IL-6 | NS | 4 |

*(Continued)*

**Table 2.** (Continued)

| Reference | Country | Study design | Number of subjects | Socio-economic status/ethnicity | Delivery mode | Assessment of depression | | Inflammatory protein markers | | | Results | Total quality score |
|---|---|---|---|---|---|---|---|---|---|---|---|---|
| | | | | | | Instruments | Timepoint(s) | Biological fluid/hour of collection | Timepoint(s) | Dosage assessment technique | Inflammatory markers | | |
| [3]* | USA | Longitudinal | T = 114 | Caucasian: 47.4%; 100% high school level or above; 36% income ≤$15,000. | NR | SCID; EPDS (risk of significant depressive symptoms—binary outcome: EPDS≥13). | 1st, 2nd, and 3rd trimester and PP | Plasma/9:00 am–12:00 am | 1st, 2nd, and 3rd trimester and PP | (1) Multiplex assay; (2) HPLC | (1) IL-1β, IL-2, IL-6, IL-8, IL-10, TNF-α (2) Kyn, Trp | ↑IL-6 levels were associated with ↑EPDS scores (p = .012) and risk of significant depressive symptoms (p = .013). ↑IL-1β levels were associated with ↑EPDS scores (p = .021) and risk of significant depressive symptoms (p = .010). IL-6, KYN and Kyn/Trp ratio in the 2nd trimester showed >95% chance of being (+) associated with EPDS scores and risk of significant depressive symptoms in the 3rd trimester. | 7 |
| [87]* | Italia | Longitudinal (from a Cohort study EDI)& | T = 110 | 89.4% high school level or above 94.8% middle-high class | 19 cesarean sections; 33 labor inductions | EPDS | 34–36 gw, 2 days PP, 3 mo PP, 12 mo PP | Serum/NR | 34–36 gw | (1) Quantikine hs ELISA kits; (2) HPLC system | (1) IL-6, (2) Trp, Kyn | Adjusting for maternal age, ↑ prenatal Kyn levels were associated with ↓ prenatal EPDS (p = .03) [↑Trp levels, ↑ IL-6 levels were associated with ↑ prenatal EPDS (p = .04) and with the change in EPDS from pregnancy to all PP time-points (ps = .04). ↓ Kyn/Trp ratio levels, ↑ IL-6 levels were associated with ↓ EPDS at delivery (p = .05) and 12 mo PP (p = .004), and with the change in EPDS from pregnancy to 12 mo PP (p = 0.048). ↑Kyn/Trp ratio levels, ↑ IL-6 levels were associated with a ↓ in EPDS from pregnancy to 3 (p = 0.03) and 12 (p = 0.014) mo PP. | 7 |
| [88] | Italia | Longitudinal (from a Cohort study EDI)& | T = 110 | 89.4% high school level or above 94.8% middle-high class | NR | EPDS | 34–36 gw; n = 89 re-evaluated at average 52 ±19.7 hours after delivery | Serum/ Morning and Afternoon | 34–36 gw; average 52 hours after delivery | Quantikine hs ELISA | IL-6, CRP | ↑ prenatal IL-6 levels with every 1-point prenatal EPDS increase (p = .04). | 7 |
| **Postpartum** | | | | | | | | | | | | | |
| [1]* | Greece | Longitudinal | T = 56 | 100% native Greek | 70.97% Caesarian section | EPDS≥11 | 1st wk (day 4) and 6th wks PP | (1) Serum (n = 56) and (2) CSF (n = 33)/NR | (1) early in labor; (2) right before epidural analgesia was infused | ELISA | IL-6 and TNF-α | (+) correlations between CSF TNF-α and IL-6 as well as serum TNF-α levels with EPDS scores in early puerperium. 1st wk PP: ↑CSF IL-6 (p = .039*); ↑Serum TNF-α (p = .055*) and ↑CSF TNF-α (p = .009*) were predictors of ↑EPDS. 6th wk PP: ↑CSF IL-6 (p = .012*) and ↑CSF TNF-α (p = .072*) were predictors of ↑EPDS. *statistical significance (p < .100) | 6 |
| [89]* | USA | Longitudinal (secondary analysis) | T = 26 | 92% white | 100% vaginal birth | CES-D≥11 (36% were identified as demonstrating symptoms of depression on 28 day PP) | 28 days PP | Urine/9:00–10:00 am (home visits) | 0 (within 24 hr of giving birth), 7, 14 and 28 days PP | ELISAs | IL-6, IL-1β | women with depressive symptoms on Day 28 had ↑ IL-1β levels on day 14 PP compared to women without symptoms of depression (p = .045) | 6 |

(*Continued*)

**Table 2.** (Continued)

| Reference | Country | Study design | Number of subjects | Socio-economic status/ethnicity | Delivery mode | Assessment of depression | | Biological fluid/hour of collection | Inflammatory protein markers | | | Results | Total quality score |
|---|---|---|---|---|---|---|---|---|---|---|---|---|---|
| | | | | | | Instruments | Timepoint(s) | | Timepoint(s) | Dosage assessment technique | Inflammatory markers | | |
| [90] (reported also in cross-sectional table) | China | Longitudinal (from Cohort study) | T = 296 (PPD = 45) | 33.8% low/middle family's socio professional category; 60.1% high school or less; 92.9% Han | 59.5% assisted delivery (vacuum extraction or cesarean delivery) | EPDS≥12 | within 6-mo after deliver | Serum/7:00–8:00 am | Within 48 hours of delivery | (1) enzyme cycling method (2) ELISA | hs-CRP, (2)IL-6 | ↑hs-CRP levels and ↑IL-6 levels in PPD group versus non-depressed (p < .0001). (+) correlation between hs-CRP and EPDS score (p = .0001) hs-CRP independent predictor of PPD (95% CI 4.96–30.12) (+) correlation between IL-6 and EPDS score (p = .0001) IL-6 independent predictor of PPD (95% CI 3.15–18.77) | 8 |
| [91]* | Sweden | Nested case-control (from BASIC cohort study) | T = 169 (Cts = 107; PPD-symptoms group = 62) | parental leave/sick leave/ unemployed: Cts = 2.8%; PPD-symptoms group = 12.9%. University/ college: Cts = 85.0%; PPD-symptoms group = 80.6% | Cesarean: Cts = 19.6%, PPD-symptoms = 19.6% | EPDS ≥ 12 and/or MINI interview (taking antidepressants was also used to identify cases) | 6 or 8 wks PP | Plasma/8:00 am-15:00 pm | Days from delivery (M ±SD): Cts = 69.5±9.7; PPD-symptoms: 67.8 ±11.1 | Multiplex extension assay | 70 of 92 inflammatory markers (21 excluded for not having normalized protein expression for >50% of the participants and 1 for technical problems) | ↑TRANCE (penalized OR = 1.20), ↑HGF (penalized OR = 1.17), ↑IL-18 (penalized OR = 1.06), ↑FGF-23 (penalized OR 1.25) and ↑CXCL1 (penalized OR = 1.11) in woman with PPD symptoms vs Ct | 6 |
| [92] | USA | Longitudinal (secondary analysis from a randomized trial) | T = 35 | NR | NR | SIGH-ADS29 | baseline (study entry), study exit (4–8 wks post-study entry) | Serum/NR | baseline (study entry), study exit (4–8 wks post-study entry) | Latex particle enhanced immunoturbidimetric assay | CRP | NS | 6 |

Note: 3HAA: 3-hydroxyanthranilic acid; 3HK: 3-hydroxykynurenine(3HK); AA: Anthranilic acid; ADA: Adenosine deaminase; AXIN1: Axin 1; BDI: Beck's Depression Inventory; BMI: Body Mass Index; CASP8: Caspase 8; CC16: Clara Cell Protein; CD: Continuous depressive group; CES-D: Center for Epidemiological Studies Depression Scale; CCL: Chemokine (C-C motif) ligand; CRP: C-reactive protein; CSF: Cerebrospinal fluid; Cts: Controls; CXCL1: C-X-C motif chemokine ligand 1; d: Day; ELISA: Enzyme-linked immunosorbent assay; EPDS: Edinburgh Postnatal Depression Scale; FGF-23: Fibroblast growth factor 23; G-CSF: Granulocyte colony-stimulating factor; GM-CSF: Granulocyte-macrophage colony-stimulating factor; gw: Gestational weeks; h: Hours; HAM-D or HDRS: Hamilton Depression Rating Scale; HGF: Hepatocyte growth factor; HPLC: High performance liquid chromatography; hs: High-sensitivity; IBS: Irritable bowel syndrome; IFN-γ: Interferon gamma; IL-: Interleukin-; IL-1β: Interleukin 1 beta; IL-1RA: Interleukin 1 receptor antagonist; IL6R: Interleukin 6 Receptor; KA: Kynurenic acid; K/T: Kynurenine/tryptophan (K/T); KYN: Kynurenine; LIF-R: Leukemia inhibitory factor-receptor; LOD: Limit of detection; LPS: Lipopolysaccharides; M: Mean; MADRS: Montgomery Asberg Depression Rating Scale; MCP-1: Monocyte chemotactic protein 1; MCP-2: Monocyte chemotactic protein; MIF: Macrophage migration inhibitory factor; MINI: The Mini International Neuropsychiatric Interview; MIP-1α: Macrophage inflammatory protein-1 alpha; MIP-1β: Macrophage inflammatory protein-1 beta; mo: Months; ND: Non-depressive group; NPX: Normalized protein expression; NR: Not reported; NS: Not significant results; PD: Postpartum depressive group; PSS: Perceived Stress Scale; SCID: Structured Clinical Interview for DSM; SDF-1α: The stromal cell-derived factor 1 alpha; sgp130: Soluble glycoprotein 130; SIGH-ADS29: The Structured Interview Guide for the Hamilton Depression Rating Scale—Atypical Depression Symptoms (SIGH-ADS29); SIGH-SAD: Structured Interview Guide for the Hamilton Depression Rating Scale—Seasonal Affective Disorders; SIRT2: Sirtiun-2; STAI: The State-Trait Anxiety Inventory; STAMPB: STAM Binding Protein; ST1A1: Sulfotransferase Family 1A Member 1u; T: Total; TG: Temporary gestational depressive group; TNF-α: Tumor necrosis factor alfa; TRANCE: Tumor necrosis factor ligand superfamily member 11; TRP: Tryptophan; vs: Versus; wks: Weeks; ZDS: Zung Depression Rating Scale; ↑: Increase; ↓: Decrease; (+): Positive; (-): Negative.

## Associations between inflammatory biomarkers and depression during pregnancy and during the postpartum period

**Antenatal depression.** Fourteen studies (4 case-control and 10 cross-sectional) exclusively assessed the association between depression and inflammatory biomarkers during pregnancy. Most of these studies were implemented during the 2nd and 3rd trimesters and used self-reported questionnaires to measure depressive symptoms / symptomatology (both using continuous scores and cut-off points) or depressive mood [93]. Additional studies also included diagnosis of depression to define the depressed group [39,45,47,94–97].

Most of the studies (11/14) found an association between inflammatory biomarkers and depression status. In general, when significant associations were found, higher levels of mostly pro-inflammatory markers (namely, CRP– 1 in 3 studies that have assessed this biomarker, IFN-γ/IL-4 ratio—1 in 4 studies, IL-1β– 2 in 5 studies, IL-1R - in the only study that assessed this molecule, IL-5–1 in 2 studies, IL-6–2 in 10 studies, IL-8–1 in 3 studies, IL-9—in the only study that have assessed this molecule, IL-12–1 in 3 studies, IL-13–1 in 2 studies, IL-23—in the only study that have assessed this molecule, IL-33—in the only study that assessed this molecule and TNF-α– 2 in 10 studies) were associated to higher depressive symptomatology or observed in depressed groups. For example, groups with higher depression symptoms demonstrated significantly higher circulating CRP levels [67] in comparison with those with lower depression symptom scores. Increased levels of CRP were also observed in a group with depression and trauma, in comparison to healthy pregnant volunteers [94]. Additionally, a positive correlation between CES-D scores and IL-1β [98] was found, as well as higher IL-1β in the CSF showing a significant association with increases odds of PND [96]. Among other markers investigated, positive correlations were also found between IFN- γ /IL-4 ratio, IL-5, IL-9, IL-12 and IL-13 and EPDS scores [99]. IL-1RA levels were significantly higher in women with high scores for depressive symptoms (CES-D>20) in comparison with women having scores less than 20 [100]. Higher cerebrospinal fluid IL-1b, IL-23 and IL-33 concentrations at pre-cesarean time were significantly associated with increased odds of perinatal depression [96]. Higher levels of TNF-α in depressed pregnant women were found when compared to controls in three studies [47,67,95]. Nevertheless, these findings were not similar across all studies. Focusing on the inflammatory biomarkers most often studied (IL-6, IL-10, IL-1β and TNF-α) inconsistent results were obtained. IL-6 was assessed in 10 of the 14 studies that explored the association between depression and inflammatory biomarkers during pregnancy, but only in two was found a significant positive association [95,98]. The same was observed for IL-1β, with no significant associations with depression in two studies [39,50], positive associations with depression symptoms or higher levels in individuals with increased odds of PND in two studies [96,98], respectively, and negative correlations with POMS-D continuous total sum score in one study [93]. For TNF-α, found at higher levels in depressed women [47,95], negative correlations were observed with POMS-D continuous total sum scores [93] and no significant associations with depressive symptoms or diagnosis of depression in seven studies [39,50,94,96,98,99,101]. None of the studies that have explored the association between IL-10 and depression found significant results [39,47,93,96,98,99]. Moreover, three studies had no significant associations between any of the investigated inflammatory biomarkers (IL-6, IL-10, CRP, IL-1β, TNF-α, and ERVWE1) and depression [39,97,101] (Table 1); one of them showing marginal p-values for higher CES-D scores as a predictor for higher levels of IL-6 and TNF- α [101].

In contrast with most of the studies in which a concrete small number of inflammatory proteins levels were determined, one study performed an immunoassay of a panel of 92 inflammatory proteins. Lower levels of 23 mostly anti-inflammatory proteins were found in women

with antenatal depression (top three: TRAIL, CSF-1, CX3CL1) and women on SSRI treatment (top three: CSF-1, CEGF-A, IL-15RA) in comparison with controls [45] (Table 1).

In respect to markers representing the tryptophan kynurenine pathway, e.g. tryptophan (TRP) and/or kynurenine (KYN) and/or the ratio KYN/TRP were assessed in ten studies: two during pregnancy [67,102]; seven involving both pregnancy and postpartum [3,69,81,82,103–105]; and one at postpartum [106]. Findings point out some mixed results, considering the assessment time point considered and the statistical associations studied. In general, higher Kyn and Kyn/Trp ratio was found in depressed groups compared to controls, both during pregnancy [3,69] and at postpartum [104]. However, these results are not consistent, with no significant associations being found between KYN and KYN/TRP and depressive symptoms [82] or even describing the association in the opposite direction [105]. Concerning TRP, findings showed no associations between TRP levels and prenatal depression [82,105].

Among other markers investigated, most indicated a pro-inflammatory immune response in association with depressive symptoms, across the perinatal period; increase of LPS [107] and neopterin levels were found [108] whereas no differences were found for ERVWE1 levels (Human Endogenous Retrovirus WEnvC7-1 Envelope Protein) between women affected by PND and healthy controls [97].

**Postpartum depression.** The methodological diversity of the eleven studies assessing the association between inflammatory biomarkers and depression during postpartum is even larger than in the studies implemented during pregnancy. The time frame varied between one week and 12 months postpartum, but most studies were conducted between 4–12 weeks postpartum. Depressive symptoms or symptomatology was analyzed in four studies [106,109–111], depressive mood in three studies [57,112,113], and diagnosis of depression in five studies. A diversity of inflammatory biomarkers assessed was also found, namely IFN-γ, IL-1α, IL-1β, IL-2, IL-4, IL-5, IL-6, IL-8, IL-10, IL12, IL-13, IL-17, IL-18, TGF-β2, TNF-α and other inflammation-related molecules.

Most of the studies (9/11) found an association between inflammation and depression postpartum, except two studies [113,114]. Increased levels of IL-6 were associated with higher depressive symptomatology in two different studies [106,109] (assessed with EPDS, SCID-5 and CES-D), although this protein was assessed in five studies. Both [106,109] were cross-sectional studies with the time point of assessment around 8 weeks postpartum. IL-8 levels were also assessed in five studies and found to be positively associated with depression symptoms in 4 different studies all with a cross-sectional design [28,106,109,115]. The study of Fransson, E and colleagues [28], interestingly assessed how depression during late pregnancy affects inflammation around childbirth; this association was only in the group of women with premature delivery (representing 42% of the total sample); the other 3 studies included later periods of assessment.

TNF-alpha, measured in four studies, was also shown to be significantly and positively associated with depression in one study [109], negatively associated with depressive symptoms when CpG-induced TNF-alpha was measured [111] and with no significant associations observed in the other two studies [106,115]. Specifically in the study from Christian et al. [109] the association between TNF-alpha and depressive mood (CES-D) was only found in a sample of African American women assessed between the 7 and 10 postpartum weeks.

Increased levels of other immunological mediators have also shown to be associated with depressive symptomatology, namely IL-2 [115], TGF-beta2 [110] and IgG [116]. On the other hand, negative associations were described in 3 studies. Two found serum INF-gamma levels to be negatively associated with depressive symptomatology between the 4–6 weeks postpartum [57,112]. Also in a cross-sectional study, Gruenberg and collaborators have shown decreased induced-levels INF-gamma and several other cytokines, such as Il-8, TNF-alpha, IL-

4, IL-5, IL-10 and IL-13, obtained from peripheral blood mononuclear cells associated with EPDS scores above 12 [111]. It is important to note that this study congregates data from women participating at 12, 24 and 36 months postpartum, so considering, albeit in a minority, participants who were assessed beyond the 12 months after birth. Decreased levels of IL-2 have also been reported to be associated with increased risk for depressive symptomatology [106].

A lower ratio of KYN and of KYN/TRP ratio is observed during the postpartum period to that during pregnancy [69] and an increase in the K/T quotient over the postpartum period [104] was found in depressed group compared to controls. KYN levels and KYN/TRP ratio were found to be related with EPDS scores during postnatal period [69], and KYN/TRP ratio with the changes in EPDS from pregnancy to 12 months postpartum [105]. Lastly, one study reported lower CC16 (considered anti-inflammatory) in women with postpartum depression [72].

**Longitudinal studies, some including predictive approaches.** Of the 31 longitudinal studies investigating the association between inflammatory biomarkers and depression over time, 4 were focused on the pregnancy period, five on the postpartum period and 22 across both pregnancy and postpartum. Various biomarkers were used for the investigations of association with depressive symptoms measured longitudinally (namely, 3HK and 3HAA, AA, CC16, CRP, GM-CSF, IFN-γ, IL-1β, IL-1RA, IL-2, IL-4, IL-5, IL-6, IL-6R, IL-7, IL-8, IL-10, IL-12, IL-13, IL-17, KA, KYN, leptin, LIF-R, LPS, MCP/CCL2, MCP-1, MIP-1α, MIP-1β, sgp130, TNF-α and TRP), although the most common findings regarded CRP, IL-6, TNF-alpha, KYN and TRP.

The longitudinal prospective studies developed during pregnancy are consistent in pointing out higher CRP [66,67,117] and TNF-α [67,117] in highly depressed groups.

Focusing on postpartum longitudinal prospective studies, a study [118] demonstrated that IL-6 and TNF-α levels at birth were predictors of symptomatology at 1 and 6 weeks postpartum. This result was also corroborated by study [119], which also found IL-6 levels at delivery (within 48h) as an independent predictor of depressive symptoms assessed 6 months postpartum. Increased levels of other immunological mediators have shown to be associated with depressive symptomatology, namely CXCL1, FGF-23, HGF, IL-18, TRANCE [91], IL-1beta [89] and CRP [120].

Among the studies focusing C-reactive protein (CRP), three showed higher CRP levels in in association to prepartum depression and lower CRP levels in postpartum depression in the same two studies [81,82] (sample overlap) and [80]. The reports for TNF-α showed significant results in 4 studies. Although three of them showed lower TNF-α in PPD group [121] or in association with lower EPDS scores [74,75], another study displays a higher TNF-α at 8–12 gestational weeks in depressed subjects vs. a healthy group, even when BMI, age and race were controlled for [77]. Decreased levels of inflammatory markers were found in a depressive symptoms group vs. controls after adjusting for multiple testing, where women with PPD presented higher plasma levels for five inflammatory markers: CXCL1, FGF-23, HGF, IL-18 and TRANCE [22]. The opposite was nevertheless true for the panel of 23 molecules considered in a study [122], where higher levels of inflammatory biomarkers were found in depressed group vs. controls; here, a different slope of change for cytokines (IL-6, CCL3, IL-15, G-CSF) was reported between depressed and non-depressed women. In turn, considering the 17 molecules assessed in the study [83] only MIP-1β showed a positive correlation with depressive symptoms.

Of the 22 longitudinal studies across pregnancy and the postpartum period, eight examined the potential predictive role of either inflammatory markers in the later depressive symptoms, or of depressive symptoms in predicting later inflammation profiles. One study showed higher depressive symptoms as a predictor for higher levels of IL-6 at mid gestation period, as well as

a significant association between increase in depressive symptoms from early to mid-gestation and IL-6 levels [64]. In the same study, higher depressive symptoms at early or mid-gestation predict higher CRP levels at midgestational period [64]. Further, another study showed that lower IL-6 levels in the 3rd trimester predicted higher EPDS scores postpartum [85]. In another study, family history of depression, third semester cortisol AUC, and third semester IL8/IL10 predicted symptoms of PPD [89]. Moreover, increased IL-6 levels were found in depressed groups vs. controls at 24–28 gestational weeks [77], both before and after delivery [123]. Additionally, higher IL-6 levels were associated with higher prenatal [3,87] and postnatal EPDS scores [87], as well as with changes in EPDS across pregnancy [87,124]. Lower prenatal Kyn levels were associated with greater depressive symptoms in late pregnancy, with prenatal Trp levels and Kyn/Trp ration moderating the association between IL-6 levels both antenatally and postpartum [87]. One study found that cytokines and tryptophan metabolites predicted depression during pregnancy and that IL-1β and IL-6 levels were associated with severity of depression symptoms during pregnancy and postpartum [3]. Centering in the longitudinal studies implemented during the postpartum period, the study conducted by Boufidou [118] showed that the TNF-alpha levels assessed in the CSF during labor significantly predicted increased depression symptoms at either 1st and 6th weeks postpartum, while serum CSF was only associated with the symptomatology at the 1st week postpartum.

## Discussion

The present systematic review explored the association between a variety of inflammatory markers and depression, in different time points from pregnancy to postpartum period. Despite the large volume of available evidence stemming from 83 studies, a combined quantitative synthesis of all eligible studies was not feasible owing to the large variability in inflammatory markers assessed, the different study designs (cross-sectional, case-control and longitudinal studies), the different windows of exposure or outcome assessment, and the methods used for assessment for both depression and inflammatory markers, making it difficult to identify a single factor that can explain the inconsistencies among studies results.

Most studies assessed cross-sectional associations, while some few tried to assess the predictive potential of inflammatory markers for depression at later time-points, or of depressive symptoms to predict later inflammatory states. Despite the high number of molecules considered in the different studies, not many prominent and consistent associations between inflammatory markers and peripartum depressive symptoms were detected, which is in line with the main conclusion of a recent systematic review, stating that despite the substantial evidence implicating dysregulated immune activity in perinatal depression, there is a little clarity regarding a consistent immune profile, especially based on analysis of circulating peripheral cytokines [32]. One could speculate, based on some published findings, that there might be different subtypes of depression in the perinatal period [125,126], which might contribute to these inconsistent findings in different settings. Importantly, associations were somewhat different when focusing at pregnancy compared to the delivery time-point and postpartum, and mainly referred to increased levels of IL-6, IL-8, CRP and TNF-α among depressed. Evidence on the association of other inflammatory markers and PPD remains more inconclusive and replication studies are needed, especially considering the low quality of the studies included in this systematic review.

Inflammatory markers are a very broad family of heterogeneous components, which have long been reported to play a significant role in the pathogenic pathways of several neurological and psychiatric diseases [127,128]. In addition, several molecules known to be activated in the inflammatory milieu or, on the other hand, having a role as inducers of an inflammatory

response, are molecules of great interest on the molecular mechanisms of depression and other psychiatric diseases.

Congruent with previous studies on non-pregnant women with depressive symptoms [82,129–131]; the most consistent finding of the present study was the significant association between elevated CRP levels and depressive symptoms during pregnancy. Pro-inflammatory markers, such as TNF-α, IL-1beta and IL-6 are released as response to stress or tissue damage, and they in turn induce the release of acute phase proteins, i.e. CRP, into the plasma. The molecular pathways through which these cytokines can impact on the development of depressive symptoms involves the dysregulation of neurotransmitter synaptic availability of monoamines such as serotonin, noradrenaline, and dopamine, as well as the metabolism of various amino acids such as tyrosine, tryptophan, phenylalanine, and glutamate [132]. Tryptophan (TRP) metabolism plays an important role in the mechanisms associated with the gut-brain axis [133]. Specifically, the kynurenine pathway (KP) is responsible for more than 90% of TRP catabolism throughout the body, with indoleamine 2,3-dioxygenase (IDO), the key metabolic enzyme, being activated in the inflammatory environment, leading to the downstream production of a variety of neuroactive compounds. The remainder of TRP is metabolized to serotonin and indole [134]. In parallel, dysregulation of TRP metabolites such as serotonin, quinolinic acid (QUIN), and kynurenic acid (KA) has been linked to depressive behavior in animal models as well as in humans. Specifically, IL-1β and TNF-α may be responsible for the induction of p38 mitogen-activated protein kinase (MAPK), which in turn can increase the expression and function of serotonin reuptake pumps, resulting in decreased serotonin synaptic availability and subsequently in depressive-like behavior in experimental animal studies [135]. Another biological mechanism that may underlie the association between inflammation and PPD onset includes the release of reactive oxygen or nitrogen species which in turn can decrease the availability of tetrahydrobiopterin (BH4), a key enzyme co-factor in monoamine synthesis [136].

The present review also highlights the large heterogeneity of results regarding the role of different inflammatory markers in depression during different time periods. This can be at least partly explained by the window of exposure; compared to delivery and postpartum, pregnancy is a period of large HPA-axis and sex-steroid hormone changes, which may explain the robust findings of the identified studies focusing on inflammatory markers during pregnancy [137]. An earlier clinical trial [123] reported that the levels of IL-6 and its receptor (IL-6R) were significantly higher during early pregnancy than before delivery, and women who developed depressive symptoms in the early puerperium had significantly higher serum IL-6 and IL-6R concentrations than those without. However, the diversity of methodological designs between these studies, in terms of biological fluid, timing of sampling, even the week of sampling, the different study designs (cross-sectional versus longitudinal, population-based vs. case-control, etc.), the ways to assess depression—depressive symptomatology vs. diagnosis -, as well as the different power of each study and diversity of ethnicities among the study populations does not allow us to consistently explain the variations observed in the results, and future studies are suggested in order to deeply explore the influence of these aspects on the association between inflammation and perinatal depression.

An important finding of the present review was the inclusion of longitudinal studies which assessed the role of inflammatory markers as predictive markers in depressive symptoms onset. In particular, six studies assessed biomarkers at some time during pregnancy in relation to depression during postpartum, showing a potential predictive role of TNF-a and IL-6 in the diagnosis of PPD [3,68,74,75,123,138]. Moreover, three studies with longitudinal design explored the predictive role of inflammatory markers assessed during delivery in relation to depression during postpartum [90,91,118]. These results are congruent with a recent review reporting the predictive value of proinflammatory cytokines in the diagnosis of PPD during

postpartum [139]. In general, although longitudinal studies suggest that the relationship between depression and inflammation is characterized by complex bidirectional associations, existing, prospective, longitudinal research designs are still poorly equipped to investigate the dynamic interplay of depression and inflammation that unfolds over a relatively short time period [33].

## Critical appraisal: Strengths and limitations

The present study acknowledges that the systematic review of PPD epidemiology, especially on the role of inflammatory markers in PPD onset, is a rather challenging field of research mainly due to the large heterogeneity of available evidence and several inherent limitations of the individual studies. First, the definition of exposure and outcome among inflammation and depression is not always straight-forward. Further, the large heterogeneity in assessment of the inflammatory markers is perhaps the most important methodological limitation of the studies. Markers of inflammation are a heterogeneous group of very different active components. The studies have often focused on different molecules, biological fluids and have even used different methods for their identification and quantification. In addition, the identified studies assessed the role of inflammatory markers in different time points, namely during pregnancy, delivery and postpartum, thus not readily allowing a quantitative synthesis of the results in the context of a meta-analysis. Further, even depression was assessed differently, with some studies using self-reports measures that often capture perinatal distress and not depression, while others used clinical instruments used in psychiatric settings to set a diagnosis of major depression. This might account for some of the inconsistencies in results between the studies.

Among other limitations, we excluded from this systematic review results of inflammatory markers related to quantification of immune cells as well as physical properties such as erythrocyte sedimentation rate (ESR). Moreover, mRNA and epigenetic studies were also excluded, as this work focused on protein level markers. In addition, it would be essential to account for confounding from medication/pharmacological treatment as well as from potential co-exposure to multiple markers or even other molecules and hormones, to delineate unbiased associations, but this has not been possible in the overwhelming majority of evidence assessed herein.

Nonetheless, beyond these limitations, the present study followed a strict pre-registered protocol and systematically reviewed all available evidence regarding the association between different inflammatory markers at the protein level and peripartum depressive symptoms. In trying to synthesize the available evidence, we also grouped the results by window of exposure and type of study, in line with previous literature suggesting that associations are different depending on whether the assessments were carried out during pregnancy or postpartum; as well as into identified studies where inflammatory markers were tested for their predictive potential, contributing to a more nuanced understanding of the bidirectional loop between immune system function and depression during the perinatal period.

## Conclusions and practical clinical implications

The present systematic review summarized the current evidence on the association of inflammatory markers and depressive symptoms during the peripartum period. Beyond potential limitations and biases, the findings of the present work provide evidence of increased pro-inflammatory markers among women with depressive symptoms. Based on the knowledge gained by the present work, we recommend future meta-analytic works to focus on the biomarkers CRP, IL-6, IL-8, IL-10, IL-1β, TNF-α, IFN-γ and the tryptophan kynurenine pathway to evaluate the direction and the strength of the association between the inflammatory

response and prenatal depression. Further, special attention should be given to the timing of evaluating the inflammatory response and depression symptoms, the methodology of assessing depression (self-reporting vs. diagnostic instruments), and methodology for analyzing biomarkers (biological fluid and dosage assessment technique). It is essential for future research to also investigate the impact of biological fluids analyzed, as well as the possible moderating role of other variables. Lastly, the longitudinal assessment for both inflammatory biomarkers and depression is crucial in order, to gain a deeper understanding of their complex association. Addressing these methodological challenges will facilitate precision medicine approaches, and thus the development of more effective interventions and support mechanisms for individuals and families affected by perinatal depression.

## Supporting information

**S1 Checklist. PRISMA 2020 checklist.**
(DOCX)

**S1 Table. Search terms.**
(DOCX)

**S2 Table. Studies reporting results between depression and inflammation as secondary data.**
(DOCX)

**S3 Table. Summary table of study results.**
(DOCX)

## Acknowledgments

The authors would like to thank Maria Karalexi, Pietro Gambadauro, Georgios Schoretsanitis and Andrea Hess Engström for help with appraisal of results, Hsing-Fen Tu for critical constructive comments and Mariana Saraiva for support on title and abstract screening.

## Author Contributions

**Conceptualization:** Anabela Silva-Fernandes, Ana Conde, Margarida Marques, Rafael A. Caparros-Gonzalez, Emma Fransson, Ana Raquel Mesquita, Bárbara Figueiredo, Alkistis Skalkidou.

**Data curation:** Anabela Silva-Fernandes, Ana Conde, Margarida Marques, Rafael A. Caparros-Gonzalez, Emma Fransson, Ana Raquel Mesquita, Bárbara Figueiredo, Alkistis Skalkidou.

**Formal analysis:** Anabela Silva-Fernandes, Ana Conde.

**Funding acquisition:** Bárbara Figueiredo, Alkistis Skalkidou.

**Investigation:** Anabela Silva-Fernandes, Ana Conde, Bárbara Figueiredo, Alkistis Skalkidou.

**Methodology:** Anabela Silva-Fernandes, Ana Conde, Margarida Marques, Rafael A. Caparros-Gonzalez, Emma Fransson, Ana Raquel Mesquita, Alkistis Skalkidou.

**Project administration:** Anabela Silva-Fernandes, Ana Conde, Margarida Marques.

**Resources:** Bárbara Figueiredo, Alkistis Skalkidou.

**Supervision:** Bárbara Figueiredo, Alkistis Skalkidou.

**Writing – original draft:** Anabela Silva-Fernandes, Ana Conde, Emma Fransson, Alkistis Skalkidou.

**Writing – review & editing:** Anabela Silva-Fernandes, Ana Conde, Margarida Marques, Rafael A. Caparros-Gonzalez, Emma Fransson, Ana Raquel Mesquita, Bárbara Figueiredo, Alkistis Skalkidou.

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
