## [Decision Letter · Decision Letter 0]

29 May 2023

PONE-D-23-00135

Inflammatory biomarkers and perinatal depression: a systematic review

PLOS ONE

Dear Dr. Skalkidou,

Thank you for submitting your manuscript to PLOS ONE. After careful consideration, we feel that it has merit but does not fully meet PLOS ONE’s publication criteria as it currently stands. Therefore, we invite you to submit a revised version of the manuscript that addresses the points raised during the review process. Please ensure that you address all the points raised by the reviewers. 

We look forward to receiving your revised manuscript.

Kind regards,

Liliana G Ciobanu

Academic Editor

PLOS ONE

"AS-F was supported by the Portuguese Foundation for Science and Technology and the Portuguese Ministry of Science, Technology and Higher Education, through the national funds, within the scope of the Transitory Disposition of the Decree No. 57/2016, of 29th of August, amended by Law No. 57/2017 of 19 July and previously through the fellowship grant SFRH/BPD/107732/2015.

This paper is part of the COST Action Riseup-PPD CA18138 and was supported by COST under COST Action Riseup-PPD CA18138 (Virtual Mobility Grant)."        

Reviewers' comments:

Reviewer's Responses to Questions

**Comments to the Author**

1. Is the manuscript technically sound, and do the data support the conclusions?

Reviewer #1: Yes

Reviewer #2: Yes

2. Has the statistical analysis been performed appropriately and rigorously? 

Reviewer #1: Yes

Reviewer #2: N/A

3. Have the authors made all data underlying the findings in their manuscript fully available?

Reviewer #1: Yes

Reviewer #2: No

4. Is the manuscript presented in an intelligible fashion and written in standard English?

Reviewer #1: Yes

Reviewer #2: Yes

5. Review Comments to the Author

Reviewer #1: I appreciate having read this article and being able to comment. This review is a well-argued paper that addresses an important and very relevant issue to clinical practice. Also, the methodology is very rigorous.

However, some minor amendments are necessary:

1. In the introduction, include more precise information on the relationship between depression and inflammatory biomarkers

2. Please provide information about Newcastle-Ottawa Scale. Why this scale has been used?

Reviewer #2: PONE-D-23-00135

Thanks for giving me the opportunity to read this paper, which systematically reviews available literature around an important topic concerning the association between inflammation and depression during the perinatal period. The authors conducted their search on three electronic databases and retrieve 3527 references. At the final stage 83 studies were included in the review and 56 were directly assessed as their primary aim was related to investigating the association between perinatal depression and inflammation. Results highlight several associations between a number of inflammatory markers and depressive symptoms either antenatally or postnatally, although high methodological heterogeneity limits comparison among studies. The topic and issues addressed by the current paper are interesting and important. One main criticisms concern the fact that a number of systematic reviews already exist on this topic (some examples below). I think it is essential that the authors explicitly state in what ways their work extends those reviews and adds to the available literature. Also, a more systematic / critical approach is needed in the results and discussion section to enable the reader to properly judge the weight of the available evidence. For example, the authors performed the quality assessment of the studies but this is not mentioned later when presenting the results.

Please find my major concerns better outlined below

1. Introduction: on page 5 it is stated that “This intensification of pro-inflammatory activity in late pregnancy co-occurs with an increase in depressive symptoms during this period, compared to lower levels in the second trimester”. I found this part confusing - an increase in levels of pro-inflammatory markers is to some extent physiological and normative across the third trimester of pregnancy, whereas greater than typical elevations have been associated with dep symptoms. I would suggest to have two different paragraph 1) concerning normative physiological changes that occur across the perinatal period in the immune system functioning 2) outline possible associations with mood symptoms.

2. Several reviews have been already conducted on this topic – some examples are:

• McCormack, C., Abuaish, S., & Monk, C. (2023). Is There an Inflammatory Profile of Perinatal Depression?. Current Psychiatry Reports, 1-16.

• Sawyer KM. The role of inflammation in the pathogenesis of perinatal depression and offspring outcomes. Brain, Behavior, & Immunity-Health. 2021;18:100390.

• Lambert M, Gressier F. Inflammatory biomarkers and postpartum depression: a

• systematic review of literature. Canadian Journal of psychiatry 2019;64(7):471-81.

• Leff-Gelman, P., Mancilla-Herrera, I., Flores-Ramos, M., Cruz-Fuentes, C., Reyes-Grajeda, J. P., García-Cuétara, M. D. P., ... & Pulido-Ascencio, D. E. (2016). The immune system and the role of inflammation in perinatal depression. Neuroscience bulletin, 32, 398-420.

• Osborne, L. M., & Monk, C. (2013). Perinatal depression—the fourth inflammatory morbidity of pregnancy?: theory and literature review. Psychoneuroendocrinology, 38(10), 1929-1952.

In this sense it needs to be clearly stated what the current review adds to the prior available work. Also, considering that a number of reviews has been already performed and there is a consistent number of potentially available studies to include why not thinking about performing a meta-analysis?

3. On page 7 it is stated: “(2) if one of the variables could be used as a predictor for the other”. Please explain.

4. Studies selection: please specify the time range considered for the “postpartum period” on page 8.

5. Data extraction (on page 9): were information concerning possible medication/pharmacological treatment also available, extracted and assessed?

6. Please describe how the quality assessment was performed, which is the score range and how it is attributed to a study.

7. Please state sample size range for the studies included

8. The results section needs to be substantially revised in an effort to summarize the available literature. Please be more specific and systematic in the description of the studies to allow the reader to judge the relatively weight of the evidence. For example: Which is the sample size range? How many studies found a positive association, how many a negative, how many a null? With which specific marker? please try to avoid unspecific term such as “most of the studies” “in general” etc.

9. Are there differences depending on: whether diagnosis of depression rather than self-reported depressive symptoms are used? Whether the samples are from high-risk or low-risk population, minority etc.? Biological fluids?

10. Please include information about the quality assessment in the results section. This is only reported in the table. Are there differences in the association reported depending on the quality score of the paper?

11. Also, studies that analyzed the TRP/KYN pathway are included in the review but this has now been mentioned in the introduction. Please include it also in the introduction to allow readers from different fields to allow what these markers are and how they are related to the inflammatory function -this appear then in the discussion section but I would suggest to introduce it right at the beginning.

12. On page 15 it is stated This study also demonstrated that IL-6 and TNF -alpha levels at birth were predictors of symptomatology at 1 and 6 weeks postpartum? – which study?

Minor points:

1. Abstract: please add the sample size range of the studies included

6. PLOS authors have the option to publish the peer review history of their article (what does this mean?). If published, this will include your full peer review and any attached files.

Reviewer #1: No

Reviewer #2: No

---

## [Author Response · Author response to Decision Letter 0]

11 Aug 2023

Comments and responses to the Editor and reviewers:

Editor and Journal requirements:

R: We thank you for this recommendation. We revised the entire manuscript to be in line with PLOS style format.

"AS-F was supported by the Portuguese Foundation for Science and Technology and the Portuguese Ministry of Science, Technology and Higher Education, through the national funds, within the scope of the Transitory Disposition of the Decree No. 57/2016, of 29th of August, amended by Law No. 57/2017 of 19 July and previously through the fellowship grant SFRH/BPD/107732/2015. This paper is part of the COST Action Riseup-PPD CA18138 and was supported by COST under COST Action Riseup-PPD CA18138 (Virtual Mobility Grant)." 

R: Dear Editor, in response to journal requirements we inform that funders had no role in the study and we have now included the statement as well as some more information in the appropriate section, the text now reading: AS-F was supported by the Portuguese Foundation for Science and Technology and the Portuguese Ministry of Science, Technology and Higher Education, through the national funds, within the scope of the Transitory Disposition of the Decree No. 57/2016, of 29th of August, amended by Law No. 57/2017 of 19 July and previously through the fellowship grant SFRH/BPD/107732/2015. AS has received funding from the Swedish Research Council, (Dnr. 2020-01965) and the Swedish Brain Foundation (FO2021-0161, Dnr. UU-PROJ 2021/332).

This paper is part of the COST Action Riseup-PPD CA18138 and was supported by COST under COST Action Riseup-PPD CA18138 (Virtual Mobility Grant). The funders had no role in study design, data collection and analysis, decision to publish, or preparation of the manuscript.

Information on the COST Action (European Cooperation in Science and Technology) are also provided in the revised manuscript, at page 26-27. Due to institutional matters, the name of the research center of the Ana Conde have changed, so we kindly ask that the institutional affiliation of this author can be changed, also corrected in the revised manuscript (title page, line 17).

Reviewer #1: I appreciate having read this article and being able to comment. This review is a well-argued paper that addresses an important and very relevant issue to clinical practice. Also, the methodology is very rigorous.

However, some minor amendments are necessary:

1. In the introduction, include more precise information on the relationship between depression and inflammatory biomarkers 

R: We thank you for this recommendation. We have updated the Introduction with more information about the relationship between depression and inflammatory biomarkers, that consists of a proxy of the immune response (please see pages 4-5 in the revised manuscript, line 97-112).

2. Please provide information about Newcastle-Ottawa Scale. Why this scale has been used? 

R: We thank you for this question. The Newcastle Ottawa scale (NOS) assesses the quality of nonrandomized studies, including case-control and cohort studies, which fits with the study designs of the studies selected for this systematic review. In this manner, we have used three different scale forms accordingly the study design: case-control studies and cohort studies (Wells, G. A., Shea, B., O’Connell, D., Peterson, J., Welch, V., Losos, M., & Tugwell, P. (2000) as well as an adapted form for cross-sectional studies (Herzog, R., Álvarez-Pasquin, M.J., Díaz, C. et al. Are healthcare workers’ intentions to vaccinate related to their knowledge, beliefs and attitudes? a systematic review. BMC Public Health 13, 154 (2013). https://doi.org/10.1186/1471-2458-13-154). This scale is widely used in this kind of work. More detailed information about this Scale was included on the page 11, line 250-258. If the Reviewer wishes us to expand this information, we would gladly do so.

Reviewer #2: PONE-D-23-00135

Thanks for giving me the opportunity to read this paper, which systematically reviews available literature around an important topic concerning the association between inflammation and depression during the perinatal period. The authors conducted their search on three electronic databases and retrieve 3527 references. At the final stage 83 studies were included in the review and 56 were directly assessed as their primary aim was related to investigating the association between perinatal depression and inflammation. Results highlight several associations between a number of inflammatory markers and depressive symptoms either antenatally or postnatally, although high methodological heterogeneity limits comparison among studies. The topic and issues addressed by the current paper are interesting and important. One main criticisms concern the fact that a number of systematic reviews already exist on this topic (some examples below). I think it is essential that the authors explicitly state in what ways their work extends those reviews and adds to the available literature. Also, a more systematic / critical approach is needed in the results and discussion section to enable the reader to properly judge the weight of the available evidence. For example, the authors performed the quality assessment of the studies but this is not mentioned later when presenting the results.

Please find my major concerns better outlined below

1. Introduction: on page 5 it is stated that “This intensification of pro-inflammatory activity in late pregnancy co-occurs with an increase in depressive symptoms during this period, compared to lower levels in the second trimester”. I found this part confusing - an increase in levels of pro-inflammatory markers is to some extent physiological and normative across the third trimester of pregnancy, whereas greater than typical elevations have been associated with dep symptoms. I would suggest to have two different paragraph 1) concerning normative physiological changes that occur across the perinatal period in the immune system functioning 2) outline possible associations with mood symptoms. 

R: We thank the reviewer for pointing out the need for clarification. We have re-written this section (please see pages 6-7, line 141-155), with the aim to highlight that although this process is a universal and physiological process during pregnancy, there could be a) individual differences in the sensitivity to this fluctuation, where some individuals react with mood symptoms with changes in inflammatory parameters and b) the level of inflammatory reactions could be further elevated in individuals with, for example, pregnancy depressive symptoms, who have been exposed to childhood traumatic events, or they might be differing by personality type.

2. Several reviews have been already conducted on this topic – some examples are:

• McCormack, C., Abuaish, S., & Monk, C. (2023). Is There an Inflammatory Profile of Perinatal Depression?. Current Psychiatry Reports, 1-16.

• Sawyer KM. The role of inflammation in the pathogenesis of perinatal depression and offspring outcomes. Brain, Behavior, & Immunity-Health. 2021;18:100390.

• Lambert M, Gressier F. Inflammatory biomarkers and postpartum depression: a systematic review of literature. Canadian Journal of psychiatry 2019;64(7):471-81.

• Leff-Gelman, P., Mancilla-Herrera, I., Flores-Ramos, M., Cruz-Fuentes, C., Reyes-Grajeda, J. P., García-Cuétara, M. D. P., ... & Pulido-Ascencio, D. E. (2016). The immune system and the role of inflammation in perinatal depression. Neuroscience bulletin, 32, 398-420.

• Osborne, L. M., & Monk, C. (2013). Perinatal depression—the fourth inflammatory morbidity of pregnancy?: theory and literature review. Psychoneuroendocrinology, 38(10), 1929-1952.

In this sense it needs to be clearly stated what the current review adds to the prior available work. Also, considering that a number of reviews has been already performed and there is a consistent number of potentially available studies to include why not thinking about performing a meta-analysis? 

R: Thank you for this very relevant question. Although several reviews have been already conducted on this topic, not all have systematically analyzed the association between inflammation and depression during pregnancy (cross-sectionally), during the postpartum (cross-sectionally) and from pregnancy to the postpartum period. Our systematic review intended to present the diversity of studies aiming to analyze associations between perinatal depression and inflammatory biomarkers with special focus both on the study design (cross-sectional, case-control and longitudinal studies) as well as, and even more importantly, on the exact period of biomarker and depression assessment (pregnancy and/or postpartum). We even included a separate note on prospective longitudinal studies, aiming to explore the possibility of a predictive role of either depression on later inflammation profiles or conversely, of inflammatory molecules on the later development of depression, which often not specifically and systematically addressed. At the end of the introduction (pages 7-8) we now specifically describe the novelty of this systematic review in relation to the other reviews named by the Reviewer (some of them already addressed in the manuscript) and updated with the references of the newly published systematic review published by McCormack, C et al. (2023) and the narrative reviews of Leff-Gelman et al. (2016) and Osborne & Monk (2013).

As stated on page 26, the diversity of assessment methods for the different inflammatory markers, for the clinical or subclinical depressive symptoms in different time points (pregnancy, delivery and postpartum) makes a quantitative synthesis of the results in the context of a meta-analysis nearly impossible. Nevertheless, some considerations about the usefulness of a meta-analysis, namely focused on specific more prominent biomarkers are now included in the discussion section, page 26, line 624-634.

 3. On page 7 it is stated: “(2) if one of the variables could be used as a predictor for the other”. Please explain. 

R: Thank you for this suggestion. We have re-formulated the study aims at the end of the introduction (please see page 8, line 190-194). The text now reads “(2) focusing on longitudinal studies, to explore the possibility of a predictive role of either depression on later inflammation profiles or conversely, of inflammatory molecules on the later development of depression.”

4. Studies selection: please specify the time range considered for the “postpartum period” on page 8.

R: Thank you for this question. We now added in the section “Studies Selection”, page 9, line 219, the time range considered for the postpartum period. The text now reads “(up to one year after delivery)”.

5. Data extraction (on page 9): were information concerning possible medication/pharmacological treatment also available, extracted and assessed? 

R: Thank you for this question. Medications/pharmacological treatment was not included as variable of interest in our data extraction. We agree that such data is of great consequence, so this missing information is now included in the limitations section (line 619, page 25). 

6. Please describe how the quality assessment was performed, which is the score range and how it is attributed to a study. 

R: Thank you for this suggestion. We add this information in the methods (page 11, line 250-258) and results (page 12-13, line 297-309).

7. Please state sample size range for the studies included 

R: Thank you for this suggestion. We add on Results section page 12, line 289-292, the sample size range for cross-sectional and case-control studies, as well as for longitudinal studies. The text now reads “according to the period involved (pregnancy or postpartum) (for details see Table 1) involving a sample size range between 10-469. Secondly, longitudinal studies are presented, with a sample size ranging from 26 to 467, with a note on whether a study included predictive analyses (for details see Table 2).)”

8. The results section needs to be substantially revised in an effort to summarize the available literature. Please be more specific and systematic in the description of the studies to allow the reader to judge the relatively weight of the evidence. For example: Which is the sample size range? How many studies found a positive association, how many a negative, how many a null? With which specific marker? please try to avoid unspecific term such as “most of the studies” “in general” etc.

R: We thank the reviewer for pointing out the need for clarification. Because of the great variability in the studies, it is a difficult task. We have now tried to summarize in the text the available literature identifying the number of biomarkers assessed in the different studies (considering the period assessed – pregnancy and/or postpartum) and the strength and direction of the association found between the inflammatory biomarkers and depression (self-reports vs. diagnoses) (please find in the text at page 13-14, line 321-330; page 14-15, line 344-360; page 16, line 394 and 398; page 17, line 417-423; page 18, line 443-447). 

We have included the new suggested summary table we have constructed to summarize the results, as Supplementary file 3. If the Editor or the Reviewer think that we should not include it, we can omit it from the manuscript.

9. Are there differences depending on: whether diagnosis of depression rather than self-reported depressive symptoms are used? Whether the samples are from high-risk or low-risk population, minority etc.? Biological fluids? 

R: Even after this new synthesis of the findings, we concluded that there is a great deal of methodological diversity and that there isn't a single identifiable factor (such as the study quality, biological fluid, timing, and format of evaluating the inflammatory response and depression, among other aspects analyzed and listed in the tables of results) that consistently enables us to explain the variations observed (page 21, line 501-502 and line 505-519; page 23, line 561-566; page 24-25, line 597-599). The Discussion section of this paper discusses this constraint and offers ideas for future research projects based on the considerations made here (please see page 26, line 624-634).

10. Please include information about the quality assessment in the results section. This is only reported in the table. Are there differences in the association reported depending on the quality score of the paper?

R: Thank you for this suggestion. A new paragraph was included in the results on the quality assessment process (page 12-13; line 297-309) including mean and standard deviation of the quality scores for each subgroup of studies regarding study design. Unfortunately, quality did not seem to influence presence or not of significant results. Even between studies with high quality, heterogenous results were found. This information was included in the Discussion section.

11. Also, studies that analyzed the TRP/KYN pathway are included in the review but this has now been mentioned in the introduction. Please include it also in the introduction to allow readers from different fields to allow what these markers are and how they are related to the inflammatory function -this appear then in the discussion section but I would suggest to introduce it right at the beginning. 

---

## [Decision Letter · Decision Letter 1]

2 Jan 2024

PONE-D-23-00135R1Inflammatory biomarkers and perinatal depression: a systematic reviewPLOS ONE

Dear Dr. Skalkidou,

Thank you for submitting your manuscript to PLOS ONE. After careful consideration, we feel that it has merit but does not fully meet PLOS ONE’s publication criteria as it currently stands. Therefore, we invite you to submit a revised version of the manuscript that addresses the points raised during the review process.

We look forward to receiving your revised manuscript.

Kind regards,

Liliana G Ciobanu

Academic Editor

PLOS ONE

Journal Requirements:

Reviewers' comments:

Reviewer's Responses to Questions

**Comments to the Author**

1. If the authors have adequately addressed your comments raised in a previous round of review and you feel that this manuscript is now acceptable for publication, you may indicate that here to bypass the “Comments to the Author” section, enter your conflict of interest statement in the “Confidential to Editor” section, and submit your "Accept" recommendation.

Reviewer #1: All comments have been addressed

Reviewer #3: All comments have been addressed

2. Is the manuscript technically sound, and do the data support the conclusions?

Reviewer #1: Yes

Reviewer #3: Partly

3. Has the statistical analysis been performed appropriately and rigorously? 

Reviewer #1: Yes

Reviewer #3: N/A

4. Have the authors made all data underlying the findings in their manuscript fully available?

Reviewer #1: Yes

Reviewer #3: Yes

5. Is the manuscript presented in an intelligible fashion and written in standard English?

Reviewer #1: Yes

Reviewer #3: Yes

6. Review Comments to the Author

Reviewer #1: All the suggested changes have been made.

I believe it will be a paper that provides relevant information for the area.

Reviewer #3: Peer Reviewer Notes

This is a systematic review on the topic of inflammatory biomarkers and perinatal depression. There were 56 studies included where the primary aim of the studies were to assess associations between depression and inflammatory biomarkers. Their results support global hypotheses that PND and inflammatory biomarkers are associated, specifically IL-6, IL-8, CRP, TNF-alpha. They assessed and identified study heterogeneity on timing (and type?) of biological sampling, as well as timing and method for PND assessments. Strengths include their inclusion of publications in multiple languages. Weaknesses include that indiscriminate inclusion of biological fluids were included in the study, without discrimination on types of fluids/proteomics; excluded experimental studies, qualitative analyses, gray literature, case reports which can be helpful in providing hypothesis directions; data extraction did not report type of biological fluids.

Major notes

• The methods are a systematic review, but per L190-194 they are asking specific questions about cross sectional associations and predictive questions about inflammation. Therefore a meta-analysis should possibly be performed and reported to answer these questions about magnitude of effects? Please, clarify or perform and report.

• The methodology and reporting appears to be more consistent with a scoping review rather than a systematic review. Consider changing the title and revising throughout the document, that the study is a scoping review.

• As a scoping review, more details on the types of studies performed and where existing knowledge gaps lie, would be informative

• I disagree with the conclusion that there is a global need to harmonize methods. Currently, since there is so much unknown about the relationships and specificity between these inflammatory markers and PND, we need “discovery science” to flesh out potential mechanisms, timing, and other key things, before systematic reviews of an outcomes-based nature are appropriate. Please consider revising the Abstract and manuscript Conclusions. Notably, the scoping review that the authors conducted are important to identifying the differences in methods and current state of understanding on the topic, which can guide future research directions.

Minor Notes

• The introduction is quite long and includes material that does not directly relate to the rationale for the study. It should be edited for brevity and focus.

Specific Notes

• L80 “While epidemiological literature…” What does this mean? There is a lot of literature on perinatal mental health that goes beyond describing what is happening or rates/incidences of certain phenomena. Please be specific or cut this clause out.

• L182-189 This is a run-on sentence. Please edit.

7. PLOS authors have the option to publish the peer review history of their article (what does this mean?). If published, this will include your full peer review and any attached files.

Reviewer #1: No

Reviewer #3: No

---

## [Author Response · Author response to Decision Letter 1]

2 Apr 2024

Thank you to Reviewer #2 for taking the time to read and support our manuscript. 

We would like to extend our special thanks to Reviewer #1 for their insightful comments and positive feedback on the changes made by the authors in response to their initial review. Their acknowledgment of our efforts and encouragement regarding the significance of our contribution to the field are immensely gratifying.

Furthermore, we are grateful to Reviewer #3 for their diligent review and additional comments on the article. Their constructive feedback has been instrumental in further refining our manuscript and ensuring its quality and clarity.

We recognize the importance of the revision process in strengthening the scholarly impact of our work. A detailed description on how we address your comments is included below. 

Comments and responses:

Reply: We thank you for this recommendation. The final draft of the manuscript was revised concerning the reference list to include all the cited papers.

Reviewer #1:

All the suggested changes have been made. I believe it will be a paper that provides relevant information for the area.

Reply: We sincerely appreciate the feedback of Reviewer #1. It's encouraging to hear that all the suggested changes have been implemented successfully. The recognition of our efforts reinforces our commitment to delivering valuable contributions to the field. We are dedicated to ensuring that our paper provides relevant and insightful information for the area. Thank you for your positive assessment.

Reviewer #3:

This is a systematic review on the topic of inflammatory biomarkers and perinatal depression. There were 56 studies included where the primary aim of the studies were to assess associations between depression and inflammatory biomarkers. Their results support global hypotheses that PND and inflammatory biomarkers are associated, specifically IL-6, IL-8, CRP, TNF-alpha. They assessed and identified study heterogeneity on timing (and type?) of biological sampling, as well as timing and method for PND assessments. Strengths include their inclusion of publications in multiple languages. Weaknesses include that indiscriminate inclusion of biological fluids were included in the study, without discrimination on types of fluids/proteomics; excluded experimental studies, qualitative analyses, gray literature, case reports which can be helpful in providing hypothesis directions; data extraction did not report type of biological fluids.

Reply: We thank the reviewer for highlighting the importance of discriminating between body fluids. We have now incorporated this information into the results, specifically in lines 297-299, which report the details already included in Table 1 and Table 2. Additionally, we have addressed this point in the discussion section, specifically in lines 576, 581-582.

Major notes

• The methods are a systematic review, but per L190-194 they are asking specific questions about cross sectional associations and predictive questions about inflammation. Therefore a meta-analysis should possibly be performed and reported to answer these questions about magnitude of effects? Please, clarify or perform and report.

Reply: Thank you for your thoughtful suggestion. We've carefully considered the possibility of performing a meta-analysis to address the specific questions raised regarding cross-sectional associations and predictive inquiries about inflammation. While we also acknowledge the potential benefits of such an analysis, and had considered it, we believe it may not be suitable for this study. As outlined in the revised objectives (L193-202), our focus is on conducting a systematic review to comprehensively synthesize the existing literature on the association between inflammation and perinatal depression. The decision not to pursue a meta-analysis is based on several factors, including the heterogeneous nature of the procedures followed in the studies we systematically reviewed, as discussed in the article (L613-624). Additionally, our primary aim is to provide a qualitative synthesis of the available evidence rather than quantifying effect sizes, which would not be realistic, as most studies have used different timepoints for measurement of exposure and outcome, and different analytical methods. We hope this clarifies our approach. 

• The methodology and reporting appears to be more consistent with a scoping review rather than a systematic review. Consider changing the title and revising throughout the document, that the study is a scoping review.

Reply: Thank you for your insightful comment. We've carefully considered your suggestion and want to emphasize that our study adheres to the guidelines established by the PRISMA 2020 for conducting systematic reviews (http://www.prisma-statement.org/). We believe we've diligently followed these guidelines and have listed them in the PRISMA 2020 Checklist. Additionally, it's important to recognize that changing our approach to a scoping review would involve shifting the focus to a different methodology, which wasn't originally considered in the protocol registered and validated by Reviewer #1. However, we understand that our previous articulation of the objectives of our systematic review may have inadvertently caused confusion for readers. Therefore, we've revised the wording of the objectives (L193-202) to make them clearer and more aligned with the methodology we have actually followed. We hope this clarification addresses any concerns, and we remain open to further feedback or suggestions.

• As a scoping review, more details on the types of studies performed and where existing knowledge gaps lie, would be informative.

Reply: Thank you for your suggestion. We appreciate your interest in gaining more insight into the types of studies performed and identifying existing knowledge gaps. However, after careful consideration and review of our methodology, we believe that our study aligns with a systematic review rather than a scoping review. We believe that our study provides valuable insights into the topic at hand, as we do discuss existing knowledge gaps and future directions. 

• I disagree with the conclusion that there is a global need to harmonize methods. Currently, since there is so much unknown about the relationships and specificity between these inflammatory markers and PND, we need "discovery science" to flesh out potential mechanisms, timing, and other key things, before systematic reviews of an outcomes-based nature are appropriate. Please consider revising the Abstract and manuscript Conclusions. Notably, the scoping review that the authors conducted are important to identifying the differences in methods and current state of understanding on the topic, which can guide future research directions.

Reply: We greatly appreciate the critical reflection provided by Reviewer #3. We partly agree with the reviewer on this matter and believe that both perspectives hold validity. It is crucial to strive for the harmonization of time points and methods to ensure comparability between studies and possibility for meta-analyses in the future. However, it is also essential to encourage the exploration of new methods and the search for new biomarkers for perinatal depression. We have now modified the Abstract (L54-64) and manuscript Conclusions (L643-649), to address the point raised by the Reviewer..

Minor Notes

• The introduction is quite long and includes material that does not directly relate to the rationale for the study. It should be edited for brevity and focus.

Reply: Thank you for your feedback. We acknowledge your observation regarding the length of the introduction and the inclusion of material that may not directly relate to the rationale for the study. We have taken your suggestion into consideration and made efforts to reduce the length of the introduction while maintaining focus on the rationale for the study and retaining the main essential ideas. We appreciate your input and believe that these revisions will enhance the clarity and focus of the Introduction.

Specific Notes

• L80 "While epidemiological literature..." What does this mean? There is a lot of literature on perinatal mental health that goes beyond describing what is happening or rates/incidences of certain phenomena. Please be specific or cut this clause out.

Reply: Thank you for your comment. We agree that the clause "While epidemiological literature..." may not be necessary and could potentially confuse the idea we are trying to convey. We appreciate your input and have accordingly removed this clause.

• L182-189 This is a run-on sentence. Please edit.

Reply: Thank you for your feedback. We appreciate your attention to detail. The run-on sentence you mentioned has been addressed, and the revised formulation can now be found at L193-202. If you have any further suggestions or concerns, please feel free to let us know.

---

## [Editor Report · Decision Letter 2]

7 May 2024

Inflammatory biomarkers and perinatal depression: a systematic review

PONE-D-23-00135R2

Dear Dr. Skalkidou,

We’re pleased to inform you that your manuscript has been judged scientifically suitable for publication and will be formally accepted for publication once it meets all outstanding technical requirements.

Kind regards,

Liliana G Ciobanu

Academic Editor

PLOS ONE

---

## [Editor Report · Acceptance letter]

21 May 2024

PONE-D-23-00135R2 

PLOS ONE

Dear Dr. Skalkidou, 

I'm pleased to inform you that your manuscript has been deemed suitable for publication in PLOS ONE. Congratulations! Your manuscript is now being handed over to our production team.

Kind regards, 

on behalf of

Dr. Liliana G Ciobanu 

Academic Editor

PLOS ONE